# Dejavu: Post-Deployment Learning for Embodied Agents via Experience Feedback

## Abstract

Embodied agents face a fundamental limitation: once deployed in real-world environments to perform specific tasks, they are unable to acquire new useful knowledge to enhance task performance. In this paper, we propose a general post-deployment learning framework called Dejavu, which employs an Experience Feedback Network (EFN) and augments the frozen Vision-Language-Action (VLA) policy with retrieved execution memories. EFN automatically identifies the most contextually successful prior action experiences and conditions action prediction on this retrieved guidance. We adopt reinforcement learning with semantic similarity rewards on EFN to ensure that the predicted actions align with past successful behaviors under current observations. During deployment, EFN continually enriches its memory with new trajectories, enabling the agent to exhibit "learning from experience" despite fixed weights. Experiments across diverse embodied tasks show that EFN significantly improves adaptability, robustness, and success rates over frozen baselines. These results highlight a promising path toward embodied agents that continually refine their behavior after deployment. We provide the code and demo on our anonymous website https://dejavu2025.github.io/.

## 1 Introduction

Embodied intelligence is an emerging paradigm in artificial intelligence, wherein an agent learns and makes decisions through physical interaction with environment (Liu et al., 2025a; Wang et al., 2024). Recently, unified Vision-Language-Action (VLA) models have achieved remarkable generalization across diverse tasks (Zitkovich et al., 2023; Shao et al., 2025; Firoozi et al., 2025; Han et al., 2026). However, these capabilities come at the cost of relying entirely on massive offline training with a fixed, unified dataset distribution (Brohan et al., 2023; Ha et al., 2024). Once deployed, the model's weights (and thus its knowledge) remain fixed, which will not be updated without retraining (Liu et al., 2024a). Indeed, while users might expect an AI robot to continue learning from new situations, the reality is that most deployed models "stop learning" upon deployment (Liu et al., 2024a).

Given this limitation, we naturally ask: do intelligent systems always need to rewrite their internal weights to improve? Humans, for instance, often tackle new problems not by learning entirely new facts, but by recalling relevant past experiences and reusing them (Andrychowicz et al., 2017; Oh et al., 2018). For example, an auto mechanic might fix a new engine issue by remembering a similar fault in another car and copying that solution. This ability to draw on episodic memories, which gives rise to the sense of "déjà vu" from having seen something similar before, does not alter core knowledge representations, yet it enables fast adaptation to new challenges by analogy to past experiences (Goyal et al., 2022). Inspired by this intuition, we ask: can an AI agent improve itself by recalling and reusing its "experiences" in a similar way? If a neural network could leverage stored memories of situations and solutions to inform its current decision-making, effectively learning from its own experience at inference time, then even a fixed-weight model could become better over time (Goyal et al., 2022). Such an agent would gain the ability to improve performance post-deployment, simply by accumulating and drawing upon new experiences, without any gradient-based re-training (Behrouz et al., 2024; Wang et al., 2025; Bagatella et al., 2025). This intriguing concept, which we term "learning from déjà vu", motivates our approach.

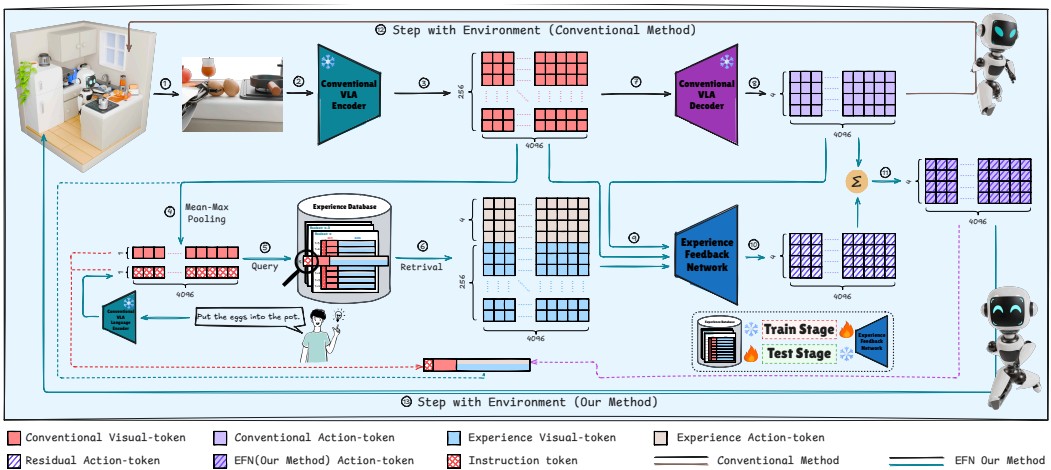

Figure 1: Top: a policy is trained once and then deployed with frozen weights, which prevents adaptation at test time. Bottom: a frozen VLA policy is augmented by an Experience Feedback Network that retrieves semantically relevant prior trajectories, produces residual corrections, and closes the loop with outcome similarity signals while keeping the base policy unchanged.

To realize this idea, we organize the design around four questions. *What is an experience, and in what format?* We define an experience as a trajectory of synchronized vision, language, and action; each rollout is stored in an experience bank aligned with the VLA interface. *How is experience reused?* We learn a network that takes the current observation together with a retrieved experience and predicts a **residual action**, which is added to the base policy's output to form the final action (Johannink et al., 2019; Liu et al., 2025b). *How is the experience matched?* Because real-world perception is dominated by vision and language, we retrieve the most relevant trajectory using language-conditioned visual similarity. *How is such a network trained?* We optimize it with reinforcement signals shaped by a dense similarity between the predicted next observation and the next state in the retrieved trajectory, enabling effective assignment while keeping the backbone frozen.

Bringing these ideas together, we introduce the Experience Feedback Network (EFN). EFN takes the current observation and an action retrieved from a matched experience, and predicts a residual action. This residual is added to the VLA policy's baseline output to produce the final control, which is then executed to yield the next observation. The intuition is straightforward: when a strong prior experience exists, EFN should exploit it to refine the action. We therefore train EFN with reinforcement learning using a dense, similarity-based reward: if the next observation resembles the *next observation in the retrieved experience*, the agent receives a higher reward, indicating that it is moving closer to that experience. This design supplies frequent shaping signals, unlike sparse success–failure rewards, and thus eases optimization. We optimize EFN with the soft actor–critic algorithm, which provides entropy regularization for robust exploration, stable value learning, and off-policy sample efficiency that enables effective reuse of stored experiences (Haarnoja et al., 2018).

During deployment, we maintain a *live* experience bank that is continuously augmented with newly successful rollouts. At every inference step, the agent retrieves similar trajectory in a joint vision–language embedding space and feeds the matched experience action to EFN alongside the current observation. EFN then predicts a residual action that refines the base VLA output as the final control. We integrate EFN with OpenVLA (Kim et al., 2024), UniVLA (Bu et al., 2025b), and the recent GO-1 (Bu et al., 2025a), and evaluate in both simulation and the real world: LIBERO for simulated tasks and the AgiBot G1 robot for physical experiments (Liu et al., 2023; Pumacay et al., 2024). Across all settings, EFN improves deployment-time performance over the base policies. We summarize our main contributions as follows:

- We introduce EFN as an *experience-centric* deployment-time mechanism that couples a live *experience bank* with a lightweight controller to improve VLA policies. By training such a network, the model can still *learn from experience* after deployment.

- We formalize an *experience* as a synchronized vision–language–action trajectory and retrieve candidates in a joint vision–language space; the retrieved step provides a matched *action prior* and its successor frame as the semantic target for guidance.

- We integrate EFN with OpenVLA, UniVLA, and GO-1, demonstrating consistent deployment-time improvements on both simulation tasks and real-world platform.

## 2 Background

### 2.1 Vision–Language–Action (VLA) Models

Large-scale VLA policies have rapidly advanced the coupling of open-vocabulary perception with end-to-end control (Shukor et al., 2025; Zhang et al., 2025; Zhai et al., 2025; Cheang et al., 2025). The Robotics Transformer family established that scaling data and model capacity yields substantial cross-task generalization in real-world manipulation (Brohan et al., 2023; Zitkovich et al., 2023). Building on this, open-source generalist policies trained on heterogeneous, multi-robot corpora demonstrated practical adaptation to new sensors and action spaces with modest finetuning, improving accessibility and reproducibility for the community (Ha et al., 2024; Kim et al., 2024).

Recent architectures emphasize *efficiency* without sacrificing reasoning ability: state-space–inspired designs reduce inference cost while preserving long-horizon context, enabling deployment on resource-constrained platforms (Liu et al., 2024b). In parallel, standardized benchmarks for compositional generalization and knowledge transfer provide clearer axes to evaluate scale, robustness, and post-deployment behavior of VLA policies (Liu et al., 2023). These developments position VLAs as strong frozen backbones that can be *augmented at inference time*—a setting where our experience feedback mechanism refines actions using retrieved trajectories without retraining the base model.

### 2.2 Post-deployment Learning and Our Perspective

A central challenge in deploying robotic policies is *improving competence after deployment* without retraining. Human-in-the-loop frameworks study how robots can collect corrective signals and update behavior during real operations, showing that runtime monitoring and continual data collection can safely enhance autonomy in the field (Liu et al., 2024a). A complementary thread reduces the burden on parametric updates by exploiting *retrieval at inference time*: retrieval-augmented reinforcement learning conditions decision-making on relevant past trajectories, so the agent can leverage experience without immediately folding it into weights (Goyal et al., 2022; Toteja et al., 2025; Bacciu et al., 2023; Tarasov et al., 2025). Orthogonal to retrieval, *residual policy* improves a strong but imperfect controller by predicting an additive correction, enabling faster adaptation than learning a policy from scratch (Johannink et al., 2019). Finally, benchmarks for *lifelong and continual robot learning* provide axes to quantify transfer, robustness, and knowledge accumulation across tasks, highlighting the need for systematic evaluation of post-deployment behavior (Liu et al., 2023).

**Our differences.** As shown in 1, EFN targets the post-deployment setting and keeps the base VLA *frozen*. Instead of updating weights online, EFN (i) retrieves a task-relevant experience trajectory, (ii) predicts a *residual action* that refines the base policy's output, and (iii) optimizes the residual via dense, similarity-shaped reinforcement signals that compare the observed next frame to the next frame in the retrieved trajectory. This design combines the benefits of retrieval (conditioning on episodic experience at test time) with residual correction (lightweight, additive refinement), yielding a practical path to deployment-time improvement without gradient-based finetuning of the underlying VLA.

### 2.3 Embodied Reinforcement Learning

Embodied reinforcement learning studies how agents acquire control policies through trial-and-error interaction with the physical or simulated world, facing challenges such as sparse rewards, sample efficiency, and robustness under real-world noise. Classic advances improved learning from sparse signals via goal relabeling (Andrychowicz et al., 2017), enabled large-scale real-robot training for vision-based manipulation (Bodnar et al., 2020), and accelerated fine-tuning by leveraging offline data before online improvement (Nair et al., 2020). Subsequent work emphasized data efficiency from pixels through strong regularization and augmentation (Yarats et al., 2021), while model-based methods demonstrated broad generality by learning world models that support imagination-based policy updates across diverse domains (Hafner et al., 2025). Benchmarks for compositional generalization and knowledge transfer (e.g., LIBERO) have provided standardized axes to evaluate continual

improvement and robustness in embodied settings (Liu et al., 2023; Yang et al., 2025; Zhang et al., 2024; Garcia et al., 2025).

**Our scope.** In contrast to training ever-larger *foundation* VLA policies, our focus is a post-deployment mechanism that augments a frozen base policy with *experience-driven* corrections. Concretely, we optimize an Experience Feedback Network (EFN) that retrieves trajectories from an experience bank and outputs a residual action to refine the base control. EFN is trained with reinforcement learning signals shaped by observation similarity, and in this work we instantiate the learner with Soft Actor–Critic (SAC) (Haarnoja et al., 2018)—thus *learning an experience-feedback module* rather than relearning a foundation VLA model.

## 3 Preliminaries

**Task setting.** We consider an embodied agent that interacts with an environment given a short natural-language instruction $\ell$ (e.g., *"put the cup on the plate"*). At discrete time $t \in \{0, 1, \dots\}$, the agent receives an *observation* $o_t$ and produces an *action* $a_t$. In our setting, the observation is a single RGB frame (optionally concatenated with proprioceptive states); we write $o_t := (I_t, \ x_t)$, where $I_t \in \mathbb{R}^{H \times W \times 3}$ is the image and $x_t$ is any low–dimensional robot state (e.g., gripper opening). The environment then transitions according to $o_{t+1} \sim \mathcal{T}(\cdot \mid o_t, a_t)$ and emits a (task-dependent) terminal signal when the rollout ends.

**VLA backbone.** A Vision–Language–Action (VLA) policy maps observations and language to an internal representation and a low-level command. We denote it by:

$$(Z_t, V_t) = \Pi_{\text{VLA}}(o_t, \ell), \qquad Z_t \in \mathbb{R}^{4 \times d}, \ V_t \in \mathbb{R}^{T \times d}, \tag{1}$$

where $V_t$ are visual tokens (spatial features) and $Z_t$ are latent action tokens (a short sequence that summarizes the intended control for the next step). A fixed decoder $D_\psi$ converts these latents to a continuous control:

$$u_t^{\text{base}} = D_\psi(Z_t, V_t). \tag{2}$$

Intuitively, equation 1 extracts "what the scene looks like" (via $V_t$) and a proposal for "what to do next" (via $Z_t$); equation 2 turns that proposal into motor commands.

**Experiences and memory.** We define a single *experience* (also called a trajectory) is the sequence

$$E = \big\{(o_0, a_0), \ (o_1, a_1), \ \dots, (o_L, \text{end})\big\}.$$

We maintain an *experience memory* $\langle E \rangle = \{E^{(1)}, E^{(2)}, \dots\}$ that stores step-level records extracted from prior rollouts (e.g., images, visual tokens, latent actions). Given the current tokens $(V_t, Z_t)$ and instruction $\ell$, a simple retriever selects a step from memory that is most relevant to the current situation:

$$j^\star = \text{retrieve}(V_t, \ell; \langle E \rangle), \qquad (V_t^E, Z_t^E, V_{t+1}^E) \leftarrow \langle E \rangle[j^\star]. \tag{3}$$

Here $V_t^E$ is the stored visual token matrix of the matched memory step, and $V_{t+1}^E$ is its immediate successor; these provide a concrete *execution prototype* for the current step.

## 4 Methodology

### 4.1 Overview of EFN

Our EFN framework is shown in Figures 2 for train stage and 3 for test stage. In the following sections, we structure the method in four parts for clarity: (1) **Experience Bank and Record Schema**, explaining how trajectories are recorded at the step level (images, visual tokens, latent actions), pooled into compact keys, and stored for fast constant-time access; (2) **Language-Conditioned Similarity and Retrieval**, which describes how we compute instruction-aware semantic similarity to select the most relevant experience step and its successor; (3) **Residual Policy Learning with SAC**, manifesting how EFN's actor predicts residual latent actions and is trained with Soft Actor–Critic under a dense, token-level similarity reward; and (4) **Deployment-Time Recall and Online Experience Growth**, which elaborates the inference stage of EFN that runs deterministically, retrieves guidance per step, and appends the most successful (or most promising) new rollouts back into the bank to enable continual improvement without updating base policy weights.

### 4.2 Experience Bank Design

**Storage schema.** We organize experiences by full *rollouts* $\tau = (s_1, a_1, \ldots, s_T, a_T)$ and insert into the bank every non-blank step $(s_t, a_t)$ encountered during data collection or deployment. Importantly, we do *not* pre-filter by success or failure; the rationale for keeping both kinds of outcomes (and how we leverage them) is discussed in the appendix. For each rollout $\tau$ we also store a fixed *instruction embedding* $\ell_\tau$ obtained by encoding the task description with the VLA's language model at the beginning of the episode. At the step level, we record three items: (i) the VLA vision-encoder features $\mathbf{F}_t \in \mathbb{R}^{L \times C}$ for frame $s_t$ (e.g., token features), (ii) a compact *key* vector $\mathbf{k}_t \in \mathbb{R}^{d_k}$ derived from $\mathbf{F}_t$ for retrieval, and (iii) the base policy's raw action $\mathbf{a}_t^{(0)}$ executed at that step. The bank therefore stores tuples $\left(\ell_\tau, \mathbf{F}_t, \mathbf{k}_t, \mathbf{a}_t^{(0)}\right)$ for all valid $t$ across all trajectories.

**Key construction and probabilistic top-$k$ retrieval.** Our key uses a *mean–max fusion with per-vector $\ell_2$ normalization*. First, $\ell_2$-normalize each token feature in $\mathbf{F}_t$ across channels. Then compute mean and max along the token dimension and normalize each result:

$$\tilde{\mathbf{F}}_t(\ell, \cdot) = \frac{\mathbf{F}_t(\ell, \cdot)}{\|\mathbf{F}_t(\ell, \cdot)\|_2 + \varepsilon}, \quad \mathbf{m}_t = \frac{\mathrm{mean}_\ell\big(\tilde{\mathbf{F}}_t(\ell, \cdot)\big)}{\big\|\mathrm{mean}_\ell\big(\tilde{\mathbf{F}}_t(\ell, \cdot)\big)\big\|_2 + \varepsilon}, \quad \mathbf{x}_t = \frac{\mathrm{max}_\ell\big(\tilde{\mathbf{F}}_t(\ell, \cdot)\big)}{\big\|\mathrm{max}_\ell\big(\tilde{\mathbf{F}}_t(\ell, \cdot)\big)\big\|_2 + \varepsilon}.$$

$$(4)$$

Then we fuse the two by an equal-weight average followed by a final normalization, yielding the key (here $d_k = C$):

$$\mathbf{k}_t = \frac{\frac{1}{2}\mathbf{m}_t + \frac{1}{2}\mathbf{x}_t}{\big\|\frac{1}{2}\mathbf{m}_t + \frac{1}{2}\mathbf{x}_t\big\|_2 + \varepsilon} \in \mathbb{R}^{d_k}. \tag{5}$$

At query time, we form a query vector $\mathbf{q}_t$ from the current frame via the same fusion, compute cosine similarities $s_i = \cos(\mathbf{q}_t, \mathbf{k}_i)$ to all keys, and select the top-$k$ indices $\mathcal{N}_k(\mathbf{q}_t)$. We then sample one key from this shortlist with a similarity-biased distribution:

$$p(i \mid \mathbf{q}_t) = \frac{\exp(s_i/\tau)}{\sum_{j \in \mathcal{N}_k(\mathbf{q}_t)} \exp(s_j/\tau)}, \qquad i \in \mathcal{N}_k(\mathbf{q}_t), \tag{6}$$

where $\tau > 0$ is a temperature. This "retrieve-then-sample" preserves exploration among near-matches while favoring the most semantically similar experiences.

### 4.3 Learning EFN with Residual Policy Optimization

**Problem setup.** EFN learns to *nudge* the base policy by recalling a relevant past experience and adjusting the current action so that the next observation resembles "what happened next" in experience. At step $t$, the inputs are: current visual features $\mathbf{F}_t$ and the base policy's action $\mathbf{a}_t^{(0)}$, together with a retrieved experience step $(\hat{\mathbf{F}}, \hat{\mathbf{a}}, \hat{\mathbf{F}}^+)$ and its rollout-level instruction embedding $\ell$ (retrieval is defined in the previous subsection). EFN's actor outputs a residual $\Delta\mathbf{a}_t$; the executed control is

$$\mathbf{a}_t = \mathbf{a}_t^{(0)} + \Delta\mathbf{a}_t. \tag{7}$$

Intuitively, $\mathbf{a}_t^{(0)}$ preserves the base policy's competence, and $\Delta\mathbf{a}_t$ is experience-informed correction.

**Semantic Match Reward.** To quantify the notion of "match the experience's next outcome," we compare the realized next observation $s_{t+1}$ with the experience's successor frame $\hat{s}^+$ at the *semantic* level. Let $\mathbf{u}(\cdot)$ be the mean–max fusion described earlier, applied to vision features to produce a unit vector. After executing $\mathbf{a}_t$, the environment yields $s_{t+1}$ with vision features $\mathbf{F}_{t+1}$. We define a dense similarity reward

$$r_t^{\mathrm{sem}} = \cos\big(\mathbf{u}(\mathbf{F}_{t+1}), \mathbf{u}(\hat{\mathbf{F}}^+)\big). \tag{8}$$

In practice we also regularize the residual magnitude to avoid destabilizing the base behavior:

$$r_t = \lambda_{\mathrm{sem}} r_t^{\mathrm{sem}} - \lambda_{\mathrm{res}} \|\Delta\mathbf{a}_t\|_2^2, \tag{9}$$

with $\lambda_{\mathrm{sem}}, \lambda_{\mathrm{res}} > 0$. This reward directly encodes our training signal without requiring supervised residual labels; a discussion on why we avoid direct supervision appears in the appendix.

**SAC objective.** We train EFN with Soft Actor–Critic, conditioning both actor and critics on the current and experience context. Let

$$\mathbf{c}_t \;=\; \text{enc}\big(\mathbf{F}_t,\, \mathbf{a}_t^{(0)},\, \hat{\mathbf{F}},\, \hat{\mathbf{a}},\, \ell\big) \tag{10}$$

be a learned context representation (the base policy is frozen). The stochastic residual policy is $\pi_\phi(\Delta\mathbf{a}_t \,|\, \mathbf{c}_t)$, and the Q-functions $Q_{\theta_1}, Q_{\theta_2}$ evaluate the corrected action $\mathbf{a}_t = \mathbf{a}_t^{(0)} + \Delta\mathbf{a}_t$ under $\mathbf{c}_t$. The critic targets use the soft Bellman backup with target networks $\bar{\theta}_i$:

$$y_t \;=\; r_t \;+\; \gamma\, \mathbb{E}_{\Delta\mathbf{a}_{t+1}\sim\pi_\phi(\cdot\,|\,\mathbf{c}_{t+1})}\Big[\min_{i=1,2} Q_{\bar{\theta}_i}\big(\mathbf{c}_{t+1},\, \mathbf{a}_{t+1}^{(0)} + \Delta\mathbf{a}_{t+1}\big) \;-\; \alpha\, \log \pi_\phi(\Delta\mathbf{a}_{t+1}\,|\,\mathbf{c}_{t+1})\Big]. \tag{11}$$

The critic loss is the standard squared error:

$$\mathcal{L}_{\text{critic}}(\theta_1, \theta_2) \;=\; \sum_{i=1,2} \mathbb{E}\Big[\big(Q_{\theta_i}(\mathbf{c}_t,\, \mathbf{a}_t^{(0)} + \Delta\mathbf{a}_t) - y_t\big)^2\Big]. \tag{12}$$

The actor minimizes the entropy-regularized objective

$$\mathcal{L}_{\text{actor}}(\phi) \;=\; \mathbb{E}_{\Delta\mathbf{a}_t\sim\pi_\phi(\cdot\,|\,\mathbf{c}_t)}\Big[\alpha\, \log\pi_\phi(\Delta\mathbf{a}_t\,|\,\mathbf{c}_t) \;-\; \min_{i=1,2} Q_{\theta_i}(\mathbf{c}_t,\, \mathbf{a}_t^{(0)} + \Delta\mathbf{a}_t)\Big], \tag{13}$$

with temperature $\alpha$ optionally tuned to maintain a target entropy. During training, gradients do not flow through the retrieval targets $\hat{\mathbf{F}}, \hat{\mathbf{F}}^+$; updates are confined to EFN's actor, critics, and the context encoder.

**Reward shaping with anti-idling penalty.** To retain absolute credit for matching the retrieved next observation while penalizing "similar but no progress" and encouraging shorter rollouts, we define semantic similarities (sim $\in [0,1]$)

$$s_t^{\text{next}} = \text{sim}(\mathbf{F}_{t+1}, \hat{\mathbf{F}}^+), \qquad s_t^{\text{cur}} = \text{sim}(\mathbf{F}_t, \hat{\mathbf{F}}), \qquad s_t^{\text{stay}} = \text{sim}(\mathbf{F}_{t+1}, \mathbf{F}_t). \tag{14}$$

We introduce auxiliary terms (with $[x]_+ = \max(x, 0)$ and tolerance $\varepsilon > 0$):

$$a_t \;=\; s_t^{\text{next}}, \qquad p_t \;=\; s_t^{\text{next}} - s_t^{\text{cur}}, \qquad m_t \;=\; 1 - s_t^{\text{stay}}, \qquad n_t \;=\; [\varepsilon - p_t]_+. \tag{15}$$

The dense reward becomes

$$r_t \;=\; w_{\text{abs}}\, a_t \;+\; w_{\text{prog}}\, [p_t]_+ \;+\; w_{\text{mot}}\, m_t \;-\; w_{\text{lazy}}\big(s_t^{\text{next}}\, n_t\, s_t^{\text{stay}}\big) \;-\; \lambda_{\text{time}}, \tag{16}$$

where $w_{\text{abs}}, w_{\text{prog}}, w_{\text{mot}}, w_{\text{lazy}} \geq 0$ and $\lambda_{\text{time}} \geq 0$ are scalars. Equation equation 16 preserves an *absolute similarity* bonus ($a_t$) so that high next-frame alignment is always rewarded; adds a *progress* term ($[p_t]_+$) to further credit genuine improvement toward the retrieved next state; encourages *non-trivial motion* ($m_t$) to avoid degenerate idling; and introduces a targeted *anti-idling penalty* ($s_t^{\text{next}} n_t s_t^{\text{stay}}$) that activates only when the prediction is already similar yet shows negligible improvement and little change between consecutive frames. The per-step cost $\lambda_{\text{time}}$ favors shorter successful trajectories. Conceptually, EFN thus learns a residual policy that steers $\mathbf{F}_{t+1}$ toward $\hat{\mathbf{F}}^+$, while avoiding "standing still to farm similarity."

4.4 Deployment-Time Retrieval and Online Experience Growth

**Goal and key differences.** At deployment, EFN recalls and reuses prior experiences while *not* updating the base policy's weights. The inference pipeline mirrors training but differs in three places. First, retrieval is *task-filtered*: we restrict matches to rollouts whose instruction embeddings are close to the current task. Second, we *prioritize efficient rollouts*: shorter trajectories receive higher selection priority because they typically contain fewer redundant actions and lead to faster completion. Third, we *grow the bank online*: after a rollout finishes, its non-blank steps are inserted into the bank so that future episodes can recall them.

**Instruction-filtered candidate set.** Given a task description, we compute an instruction embedding with the VLA's language encoder, denoted $\ell^\star$. We compare $\ell^\star$ to all stored rollout-level embeddings $\{\ell_{\tau_j}\}$ with cosine similarity and select the top-$n$ rollouts:

$$\mathcal{R}_n \;=\; \text{Top-}n\,\big\{\cos(\ell^\star, \ell_{\tau_j})\big\}_j. \tag{17}$$

All step-level entries from these rollouts form the *candidate experience set* $\mathcal{C}$, which is the only pool we retrieve from during this episode.

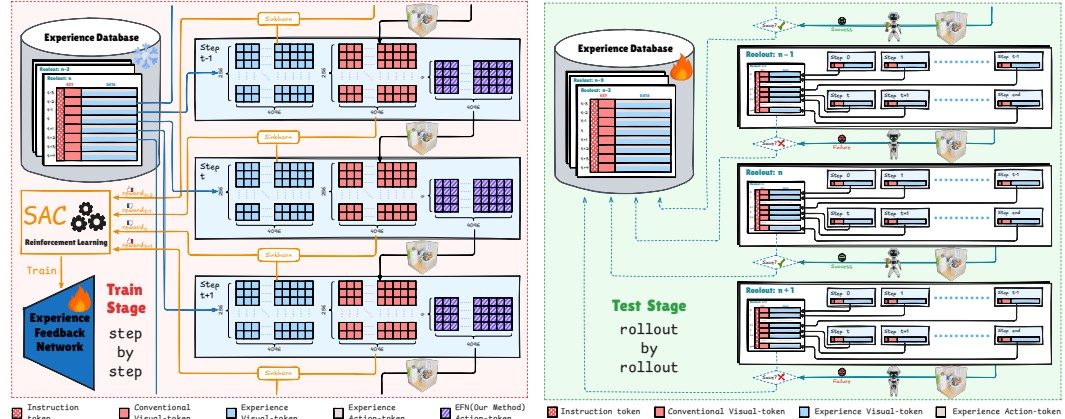

Figure 2: EFN trains a residual policy with SAC to nudge the base action so the next frame matches the retrieved memory's successor.

Figure 3: At inference, EFN filters memories by instruction, retrieves efficient candidates, applies the residual correction and grows the bank online.

**Step-wise retrieval with efficiency prior.** At step $t$, we form a visual query $\mathbf{q}_t = \mathbf{u}(\mathbf{F}_t)$ using the mean–max fusion $\mathbf{u}(\cdot)$ defined previously. Each candidate $i \in \mathcal{C}$ has a key $\mathbf{k}_i$, a successor feature $\hat{\mathbf{F}}_i^+$, and belongs to a rollout $\rho(i)$ with total length $L_{\rho(i)}$. We score candidates by combining semantic similarity and an efficiency prior that favors shorter rollouts. Let $s_i = \cos(\mathbf{q}_t, \mathbf{k}_i)$ and define a normalized length prior $g(L_{\rho(i)}) \in [0, 1]$ that decreases with $L_{\rho(i)}$ (e.g., $g(L) = \exp[-\beta L/\bar{L}]$ with temperature $\beta$ and reference length $\bar{L}$). The combined score is

$$\tilde{s}_i = \lambda s_i + (1 - \lambda) g(L_{\rho(i)}), \qquad \lambda \in [0, 1]. \tag{18}$$

We take the top-$k$ candidates by $\tilde{s}_i$ and sample one with softmax:

$$p(i \mid t) = \frac{\exp(\tilde{s}_i/\tau)}{\sum_{j \in \mathcal{N}_k(t)} \exp(\tilde{s}_j/\tau)}, \qquad i \in \mathcal{N}_k(t), \tag{19}$$

where $\tau$ is a temperature and $\mathcal{N}_k(t)$ denotes the $k$ highest-scoring items at step $t$. This procedure preserves exploration among near-matches while preferring memories that both look similar and come from efficient behaviors.

**Action correction and execution.** Conditioned on the current context and the sampled experience $(\hat{\mathbf{F}}_i, \hat{\mathbf{a}}_i, \hat{\mathbf{F}}_i^+)$, EFN predicts a residual $\Delta\mathbf{a}_t$ and executes $\mathbf{a}_t = \mathbf{a}_t^{(0)} + \Delta\mathbf{a}_t$ as in training. All critics and the policy remain fixed at inference; if desired, we use a deterministic mean action or a low-temperature sample to reduce variance. The semantic objective from training carries over conceptually: the correction is chosen to steer the next observation toward the stored successor $\hat{\mathbf{F}}_i^+$.

**Online experience growth.** After the episode ends, we insert the new rollout into the bank: we store the episode-level instruction embedding $\ell^\star$ with all non-blank step tuples $(\mathbf{F}_t, \mathbf{k}_t, \mathbf{a}_t^{(0)})$ computed with the same mean–max keying. Consistent with our storage policy, we do not filter by success or failure; reasons and ablations are provided in the appendix. In practice, when operating under a experience budget, one can apply standard retention strategies (e.g., reservoir-style sampling or recency-aware replacement) without changing the retrieval or learning rules described above.

## 5 Experiments

**Experimental Setup** We evaluate EFN in the LIBERO simulator on the `libero_goal` benchmark (Liu et al., 2023). The visuomotor backbone is a pretrained OpenVLA policy executed in bfloat16 with Flash Attention (Dao, 2023). Visual inputs follow the Prismatic pretraining pipeline (center crop; resize to 256). Each episode starts with a 10-step settling period without control. Uni-VLA provides a sequence of 256 visual tokens and 4 latent action tokens per step over a window of length 12. EFN receives the current tokens and those retrieved from the experience bank; the actor

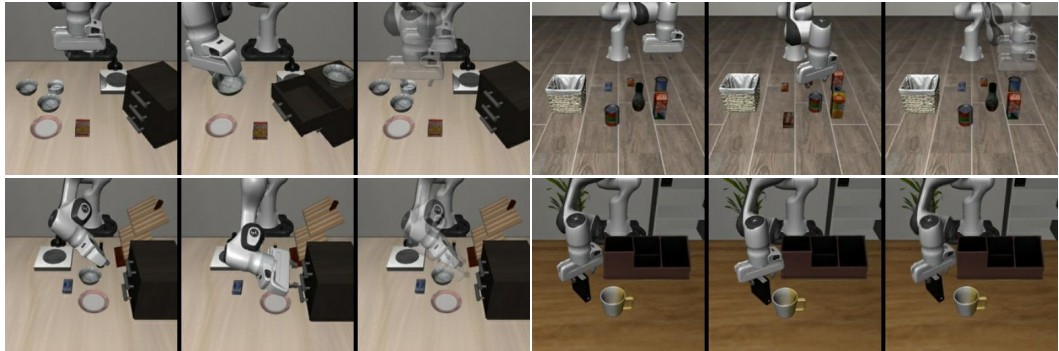

Figure 4: Visualization of EFN's residual action: corrections become smaller and more targeted as the bank grows. Within each patch: Left: Original Observation; Middle: Experience Observation; Right: Corrected action vs original action.

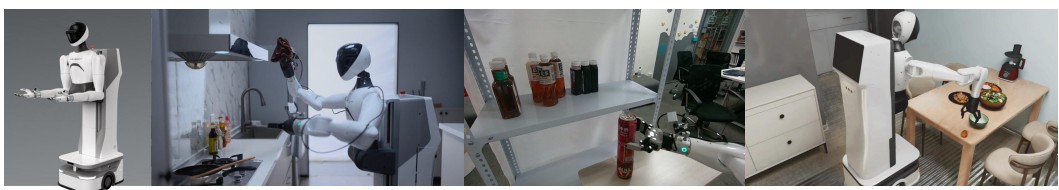

Figure 5: AgiBot-G1 platform used in our real-world experiments.

predicts a residual in latent space that is added to the base latent action and decoded by the *frozen* action head. The detailed training protocol can be found in Appendix B.

**Evaluation protocol.** At test time the policy is deterministic by applying tanh to the actor mean in token space. We report success rate and, conditional on success, the average number of steps (lower is better). For `libero_goal` the horizon cap is $H=320$. The evaluation-time experience bank mirrors training and is queried via cosine nearest neighbors over pooled visual embeddings. After each successful rollout, we append the episode (images, token features, pooled keys) to the bank; among successes, shorter episodes are retained with higher priority to efficiency.

Table 1: Deployment Performance on Libero Dataset by OpenVLA and UniVLA

| Method | Spatial | | Object | | Goal | | Long | | Average | |
| --- | --- | --- | --- | --- | --- | --- | --- | --- | --- | --- |
| | Succ. ↑ | Step ↓ | Succ. ↑ | Step ↓ | Succ. ↑ | Step ↓ | Succ. ↑ | Step ↓ | Succ. ↑ | Step ↓ |
| OpenVLA (Kim et al., 2024) | 84.7 | 119.5 | 88.4 | 163.7 | 79.2 | 121.5 | 53.7 | 275.9 | 76.5 | 160.2 |
| +EFN(Volume=100) | 86.2 | 117.0 | 90.1 | 161.0 | 81.4 | 119.4 | 64.8 | 270.6 | 80.6 | 160.8 |
| +EFN(Volume=300) | 88.5 | 115.4 | 91.3 | 158.8 | 85.7 | 117.0 | 72.1 | 267.2 | 84.4 | 160.0 |
| +EFN(Volume=500) | **90.2** | 111.8 | 92.0 | 158.3 | 88.1 | **114.5** | 75.7 | 264.3 | 86.5 | 158.2 |
| +EFN(Volume=1000) | 89.9 | **109.0** | **92.2** | **156.1** | **89.2** | 115.2 | **76.5** | **261.9** | **87.0** | **156.7** |
| UniVLA (Bu et al., 2025b) | 96.5 | 112.7 | 96.8 | 159.0 | 95.6 | 124.9 | 92.0 | 264.5 | 95.2 | 164.2 |
| +EFN(Volume=100) | 97.2 | 107.6 | 97.4 | 154.9 | 96.4 | 122.2 | 92.9 | 258.1 | 96.0 | 159.7 |
| +EFN(Volume=300) | 97.7 | 103.4 | 97.9 | 151.4 | 97.2 | 120.1 | 93.7 | 253.8 | 96.6 | 156.2 |
| +EFN(Volume=500) | 98.1 | **101.8** | **98.2** | 146.7 | 97.4 | **117.6** | 94.3 | 244.1 | 97.0 | 151.7 |
| +EFN(Volume=1000) | **98.2** | 102.1 | **98.2** | **145.8** | **97.6** | 117.8 | **94.6** | **242.5** | **97.2** | **151.3** |

**Baselines and variants.** We compare OpenVLA and UniVLA backbones with and without EFN, and study the effect of bank capacity (`Volume` $\in \{100, 300, 500, 1000\}$). Real-world tests are conducted on the AgiBot-G1 platform with the GO-1 policy (Bu et al., 2025a).

**Results on LIBERO.** Tables (OpenVLA/UniVLA) show that adding EFN consistently improves average success while reducing steps, and larger banks yield further gains. Improvements are most pronounced on the `Long` split, indicating that recall-guided residuals help truncate redundant behavior. UniVLA starts strong and still benefits from EFN, suggesting complementary value between

Table 2: Real-World Experiment of AgiBot G1 Robot with GO-1 model on three tasks

| Benchmark | PutBottle | | SortItem | | AddGoods | |
|---|---|---|---|---|---|---|
| | Succ. ↑ | Step ↓ | Succ. ↑ | Step ↓ | Succ. ↑ | Step ↓ |
| GO-1 (Bu et al., 2025a) | 46.9 | 411.0 | 34.3 | 443.5 | 15.6 | 388.0 |
| +EFN(Volume=50) | 56.3 | 398.2 | 40.6 | 427.8 | 28.1 | 372.1 |
| +EFN(Volume=100) | 65.6 | 386.8 | 43.8 | 419.9 | 34.4 | 364.4 |
| +EFN(Volume=300) | **68.8** | 383.5 | 46.9 | 416.0 | 40.6 | 359.2 |
| +EFN(Volume=500) | **68.8** | **379.4** | **50.0** | **414.3** | **43.8** | **358.0** |

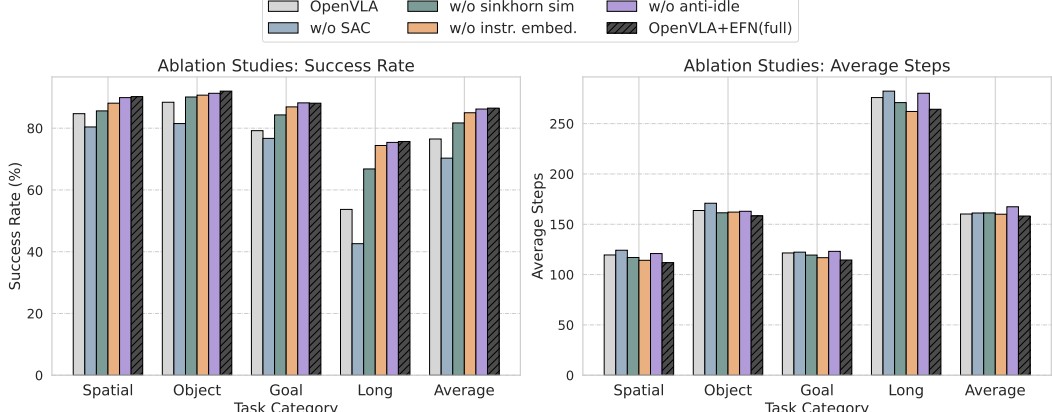

Figure 6: Ablation studies of EFN on Libero Dataset.

a strong backbone and experience-guided correction. Figure 1 visualizes residual actions; qualitatively, EFN produces smaller, more purposeful adjustments as the bank grows. We show the performance of the predicted residual actions in Figure 4.

**Real-world results.** On AgiBot-G1 (Figure 5) across three manipulation tasks, EFN boosts success and shortens trajectories as bank size increases, with diminishing returns beyond a few hundred entries, as shown in Table 2. This aligns with the simulator trend and indicates that EFN's retrieval-and-correct mechanism transfers to physical systems without changing the frozen backbone. The detailed instructions of these tasks is in Appendix B.

**Ablations.** Our evaluation of four variants validates our design choices. The w/o SAC variant, which replaced SAC with a simpler critic, degraded both success and efficiency, confirming the importance of entropy-regularized optimization. Similarly, the w/o sinkhorn sim variant, using cosine similarity instead of our Sinkhorn OT reward, provided a weaker training signal and lower performance. Removing instruction-based filtering in the w/o instruction embed variant led to retrieval mismatches and consistently underperformed, while dropping the penalty in the w/o anti-idle model increased dithering and average steps. Our conclusion is that these ablations prove each component is critical, with the complete EFN achieving the best overall balance of success and efficiency.

## 6 Conclusion

We introduced the Experience Feedback Network (EFN), which augments a frozen vision–language–action policy with a residual controller and episodic memory. By retrieving semantically relevant trajectories and imitating their next transitions through token-level optimal transport, EFN enables deployment-time adaptation without modifying pretrained weights. This transforms occasional successes into reliable performance and reframes post-deployment learning as case-based control: the backbone provides competence, memory provides context, and the residual head provides adaptation. With advances in memory, retrieval, and credit assignment, experience-driven adaptation offers a promising path to bridge offline generalization and reliable on-site execution for embodied agents.

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

# Appendix

## A  Discussion

### A.1  Limitations

The proposed framework inherits a set of practical and conceptual limitations. The first limitation concerns memory growth and curation. The experience memory expands as deployment proceeds, which increases retrieval latency and the risk of recalling suboptimal or redundant prototypes. The current policy mitigates this by prioritizing successful and shorter episodes within task buckets; however, this heuristic does not guarantee global optimality and may discard rare but informative failures. A second limitation is retrieval ambiguity. When the instruction is broad or when multiple experiences are semantically close, the nearest–neighbour criterion can select a trajectory whose next state is misaligned with the current phase, which introduces non–stationary targets for the critic and may slow policy improvement. The third limitation is the cold–start condition. If the bank contains no success and only weak partial progress, the agent may imitate unhelpful behaviours, which delays the discovery of the first success that bootstraps reliable performance. The fourth limitation lies in the reliance on visual token alignment. The imitation reward assumes that the next observation from the current attempt can be meaningfully matched to the next observation of the retrieved experience; strong viewpoint or lighting shifts can degrade the signal, and while entropic optimal transport provides robustness, it adds computational overhead. The fifth limitation is stability of residual control. Large residuals in latent space may push the decoder into regimes insufficiently covered by pretraining; the tanh squashing and action clipping provide safeguards, yet the residual head may still overshoot when the retrieved prototype poorly represents the present scene. Finally, the method trades statistical updates of the frozen backbone for case–based recall; this avoids catastrophic forgetting but also constrains asymptotic optimality when truly new skills beyond the backbone's representational capacity are required.

### A.2  Future directions

Several extensions can address the above limitations. Memory can be made adaptive through online clustering with diversity–aware reservoir sampling that preserves both canonical successes and atypical but useful corner cases. Retrieval can be learned end–to–end with a contrastive objective that aligns the query embedding of the current state with embeddings of experiences that yielded high downstream imitation rewards, while repelling confounding near–misses; a learned temperature can emphasize discriminative dimensions for disambiguation. Phase awareness can be strengthened by aligning short sub–segments instead of single steps, using dynamic time warping in the token space to stabilize the reference index. The residual policy can incorporate uncertainty estimates to attenuate residual magnitudes when retrieval confidence is low, shifting execution weight back to the frozen backbone until a confident prototype is found. Long–horizon tasks can benefit from compositional recall that stitches segments from multiple experiences into a temporally consistent pseudo–plan, with consistency enforced by overlap constraints in token space. The Sinkhorn computation can be accelerated by low–rank kernel approximations and token pruning schedules that keep only salient patches identified by language–conditioned attention. Finally, safety and interpretability can be improved by attributing which retrieved tokens and which residual channels most affected the decoded action, enabling human oversight and selective memory editing.

Table 3: Ablation Studies on Libero Dataset

| Method | Spatial | | Object | | Goal | | Long | | Average | |
|---|---|---|---|---|---|---|---|---|---|---|
| | Succ. | Step | Succ. | Step | Succ. | Step | Succ. | Step | Succ. | Step |
| OpenVLA | 84.7 | 119.5 | 88.4 | 163.7 | 79.2 | 121.5 | 53.7 | 275.9 | 76.5 | 160.2 |
| w/o SAC | 80.4 | 124.2 | 81.5 | 170.9 | 76.7 | 122.3 | 42.6 | 282.3 | 70.3 | 161.2 |
| w/o sinkhorn sim | 85.6 | 117.0 | 90.1 | 161.4 | 84.3 | 119.4 | 66.8 | 270.9 | 81.7 | 161.3 |
| w/o instruction embed | 88.1 | 114.2 | 90.7 | 162.1 | 86.9 | 116.8 | 74.4 | **262.1** | 85.0 | 160.0 |
| w/o anti-idle | 89.9 | 120.9 | 91.3 | 162.9 | **88.2** | 123.1 | 75.4 | 280.2 | 86.2 | 167.4 |
| OpenVLA+EFN(full) | **90.2** | **111.8** | **92.0** | **158.3** | 88.1 | **114.5** | **75.7** | 264.3 | **86.5** | **158.2** |

## B Training Protocal

We train EFN with goal-conditioned Soft Actor–Critic (SAC). The actor is a lightweight transformer with cross-attention over current and retrieved tokens (embed dim 1024, 16 heads, FFN size 4096, 2 layers with residual connections and LayerNorm). The twin critics pool visual tokens with a multi-latent attention block and project current latent, retrieved latent, and residual into a shared 512-d space before an MLP head (hidden 1024). The replay buffer stores $(x_t, \tilde{x}_t, y_t, r_t, d_t)$, where $x_t$ are current tokens, $\tilde{x}_t$ retrieved tokens, $y_t$ the residual target (reparameterized via the actor), $r_t$ the token-level optimal-transport reward, and $d_t$ the termination flag. We use batch size 32, discount $\gamma=0.98$, target update $\tau=0.005$, and Adam (Kingma & Ba, 2017) with learning rate $3\times10^{-4}$ for actor, critics, and temperature; the target entropy is set to a quarter of the residual degrees of freedom. To reduce target drift, the retrieved reference within an episode can be frozen to a stepwise-advanced index. The Sinkhorn-based similarity uses entropic regularization $\varepsilon=0.05$ and 50 iterations; the reward is linearly mapped

$$R_t \;=\; \alpha\big(\mathrm{sinkhorn}(X_t, \tilde{X}_t) \;-\; \beta\big), \tag{20}$$

with default $\alpha=1, \beta=0$.

**Specific Task Description**  The prompts of the three tasks are as follow: PutBottle: Grasp the drink bottle from the shelf using your right arm.Place the bottle next to similar items on the shelf and release the right arm gripper. SortItem: Pick up the drink bottle from the small tabletop with right arm. Place the drink bottles on the shelf with your right arm. AddGoods: Grasp the water bottle with your right arm.Place the water bottle in the box with your right arm.

## C Why We Do Not Use Supervised Learning

**Non-differentiable environment dynamics.**  EFN predicts a residual $\mathbf{r}_\theta$ that is *added* to the base action $\mathbf{a}_t$ and then *executed* in the environment:

$$\mathbf{a}_t^{\mathrm{efn}} \;=\; \mathbf{a}_t + \mathbf{r}_\theta\big(\underbrace{\mathbf{o}_t, \mathbf{a}_t}_{\text{current}}, \underbrace{\tilde{\mathbf{o}}_t, \tilde{\mathbf{a}}_t}_{\text{retrieved}}\big), \qquad \mathbf{o}_{t+1} \;\sim\; \mathcal{T}\big(\mathbf{o}_t, \mathbf{a}_t^{\mathrm{efn}}\big), \tag{21}$$

where $\mathcal{T}$ is the (stochastic) transition kernel of the *real* environment. Our learning signal measures how close the realized next observation is to the memory's next observation, e.g.,

$$\ell\big(\mathbf{o}_{t+1}, \tilde{\mathbf{o}}_{t+1}\big) \;=\; 1 - \mathrm{sim}\big(f(\mathbf{o}_{t+1}), f(\tilde{\mathbf{o}}_{t+1})\big), \tag{22}$$

with $f(\cdot)$ a frozen visual encoder and sim a cosine (or token-wise) similarity. A naive supervised objective would minimize $\ell$ w.r.t. $\theta$ by backpropagating through the execution in equation 21. However, this requires the Jacobian $\frac{\partial \mathbf{o}_{t+1}}{\partial \mathbf{a}_t^{\mathrm{efn}}}$ and, via the chain rule,

$$\frac{\partial \ell}{\partial \theta} \;=\; \frac{\partial \ell}{\partial \mathbf{o}_{t+1}} \cdot \underbrace{\frac{\partial \mathbf{o}_{t+1}}{\partial \mathbf{a}_t^{\mathrm{efn}}}}_{\text{env. dynamics}} \cdot \frac{\partial \mathbf{a}_t^{\mathrm{efn}}}{\partial \theta}. \tag{23}$$

In physical systems (and most simulators we rely on at train time), $\mathcal{T}$ is a black box with contact, saturation, and sensor quantization; the derivative $\partial \mathbf{o}_{t+1}/\partial \mathbf{a}_t^{\mathrm{efn}}$ is undefined or prohibitively noisy. Consequently, gradients in equation 23 are unavailable, and supervised backpropagation to $\theta$ becomes infeasible.

**Discrete retrieval and credit assignment.**  EFN conditions on a *retrieved* memory step chosen by a top-$k$ / argmax rule

$$(\tilde{\mathbf{o}}_t, \tilde{\mathbf{a}}_t, \tilde{\mathbf{o}}_{t+1}) \;=\; \arg\underset{i}{\text{top-k}} \; \mathrm{score}\big(g(\mathbf{o}_t, \text{text}), g(\mathbf{o}^i, \text{text}^i)\big), \tag{24}$$

which is a discrete operation. Even if $\mathcal{T}$ were differentiable, the retrieval introduces another non-differentiable node, further breaking end-to-end supervised learning. Moreover, supervised targets for residuals are *not identifiable*: many different residuals can lead to next observations that are semantically close to $\tilde{\mathbf{o}}_{t+1}$, so a single "ground-truth residual" label does not exist.

**RL objective circumvents the need for env gradients.** We therefore pose learning as entropy-regularized RL over continuous actions, optimizing the expected semantic reward

$$r(\mathbf{o}_t, \mathbf{a}_t^{\text{efn}}) \;=\; \text{sim}\big(f(\mathbf{o}_{t+1}),\, f(\tilde{\mathbf{o}}_{t+1})\big), \qquad J(\theta) = \mathbb{E}\Big[\sum_t \gamma^t\, r(\mathbf{o}_t, \mathbf{a}_t^{\text{efn}})\Big], \tag{24}$$

and training EFN's residual policy with Soft Actor–Critic (SAC). Using the reparameterization $\mathbf{a}_t^{\text{efn}} = \mu_\theta(\mathbf{s}_t) + \sigma_\theta(\mathbf{s}_t) \odot \boldsymbol{\epsilon}$, we maximize

$$\mathbb{E}\big[\, Q_\phi(\mathbf{s}_t, \mathbf{a}_t^{\text{efn}}) \;-\; \alpha \log \pi_\theta(\mathbf{a}_t^{\text{efn}} \mid \mathbf{s}_t)\,\big], \quad \mathbf{s}_t = \big(\mathbf{o}_t, \mathbf{a}_t, \tilde{\mathbf{o}}_t, \tilde{\mathbf{a}}_t\big), \tag{25}$$

where $Q_\phi$ is learned off-policy. Critically, policy gradients estimate $\nabla_\theta J$ from *sampled rollouts* and do not require differentiating through $\mathcal{T}$ or the discrete retrieval. This avoids the gradient discontinuities of supervised training, provides proper temporal credit assignment, and remains stable under real-world non-smooth dynamics.

**Practical remarks.** (1) Differentiable simulators could, in principle, provide $\partial \mathbf{o}_{t+1}/\partial \mathbf{a}$, but model mismatch and contact non-smoothness introduce severe bias; learned world-models add compounding error. (2) Labeling residuals from logged data is unreliable because the base state $\mathbf{o}_t$ rarely matches the retrieved memory state $\tilde{\mathbf{o}}_t$ exactly; small state mismatch produces large label noise. (3) Off-policy SAC lets us reuse experience efficiently while shaping dense rewards from equation 22, achieving sample-efficient training without supervised targets.

## D  Language conditioned image semantic similarity

This appendix details a training free procedure for computing the semantic proximity between two visual observations under a short natural language instruction. The method is designed to be plug and play inside an experience retrieval loop and to be both storage efficient and computationally light at scale.

### D.1  Problem statement and notation

Let an observation $o$ consist of an RGB image $I$ and optional side signals. In the vision encoder, a single frame produces a matrix of token features

$$V \in \mathbb{R}^{T \times H}, \qquad T = 256,\; H = 4096, \tag{26}$$

where rows index spatial tokens and columns index feature channels. Given two observations $o_1, o_2$ and a short instruction $L$ such as *put the cup on the plate*, the objective is a scalar similarity $S(o_1, o_2 \mid L) \in [0, 1]$ that is high only when both observations depict the same instruction specific semantics. The procedure follows a coarse to fine decomposition

$$S(o_1, o_2 \mid L) = S_{\text{embed}}(o_1, o_2) \oplus S_{\text{image}}(o_1, o_2 \mid L) \oplus S_{\text{act}}(o_1, o_2), \tag{27}$$

with a fusion operator specified later.

### D.2  Token pooling on the spatial axis

Each column of $V$ encodes a channel in the learned semantic basis. Each row encodes a spatial token that captures local appearance and relations. A global summary that preserves the semantic basis should remove spatial redundancy while keeping the channel space intact. Pooling across the token axis achieves that outcome. Pooling across channels instead would collapse the learned basis and produce a length $T$ vector of patch magnitudes with poor semantic fidelity.

The spatial pooling is defined by per token normalization followed by mean and max aggregation

$$\tilde{V}_{i:} = \frac{V_{i:}}{\|V_{i:}\|_2 + \varepsilon}, \qquad m = \frac{1}{T}\sum_{i=1}^{T} \tilde{V}_{i:}, \qquad x = \max_{1 \le i \le T} \tilde{V}_{i:}, \tag{28}$$

$$u_0 = \frac{\frac{1}{2}\frac{m}{\|m\|_2 + \varepsilon} + \frac{1}{2}\frac{x}{\|x\|_2 + \varepsilon}}{\|\frac{1}{2}\frac{m}{\|m\|_2 + \varepsilon} + \frac{1}{2}\frac{x}{\|x\|_2 + \varepsilon}\|_2 + \varepsilon} \in \mathbb{R}^H. \tag{29}$$

The vector $u_0$ is a global semantic descriptor in the model feature space.

**Random projection and quantization for storage efficiency**  To reduce storage while preserving cosine geometry, apply a fixed Johnson Lindenstrauss projection $P \in \mathbb{R}^{H \times d}$ with $d \ll H$ drawn once with a fixed seed and stored alongside the dataset

$$u = \frac{u_0 P}{\|u_0 P\|_2 + \varepsilon} \in \mathbb{R}^d. \tag{30}$$

A per vector symmetric quantizer stores $u$ as $(q, s)$ with

$$s = \frac{\max_j |u_j|}{127} + \delta, \qquad q_j = \text{round}\left(\frac{u_j}{s}\right) \in \{-127, \dots, 127\}, \tag{31}$$

and dequantization $\hat{u} = s\,q$. With $d = 256$ this yields 256 bytes for $q$ plus 4 bytes for $s$ per step before compression. For two hundred thousand steps the footprint is roughly fifty to sixty megabytes.

**Coarse similarity**  The coarse similarity is the cosine between dequantized summaries

$$S_{\text{embed}}(o_1, o_2) = \tfrac{1}{2}\left(1 + \frac{\hat{u}_1^\top \hat{u}_2}{\|\hat{u}_1\|_2 \|\hat{u}_2\|_2}\right). \tag{32}$$

D.3   Language aware image reranking

Let $z_1, z_2 \in \mathbb{R}^{d_z}$ be image features from a contrastive vision language encoder and let $t \in \mathbb{R}^{d_z}$ be the text feature of $L$. Define three components. The appearance proximity

$$S_{\text{clip}} = \tfrac{1}{2}\left(1 + \frac{z_1^\top z_2}{\|z_1\|_2 \|z_2\|_2}\right). \tag{33}$$

The instruction gate that is stringent on both images

$$S_{\text{text}} = \min\left\{\tfrac{1}{2}\left(1 + \frac{z_1^\top t}{\|z_1\|_2 \|t\|_2}\right), \ \tfrac{1}{2}\left(1 + \frac{z_2^\top t}{\|z_2\|_2 \|t\|_2}\right)\right\}. \tag{34}$$

The relation consistency that compares geometric and contact attributes extracted under $L$. Let $r_i \in \mathbb{R}^K$ collect proximity, intersection over union, contact proxy and gripper opening when available. Define

$$S_{\text{rel}} = \exp\left(-\sum_{k=1}^{K} \frac{(r_{1k} - r_{2k})^2}{2\sigma_k^2}\right). \tag{35}$$

The reranking score is a convex combination

$$S_{\text{image}} = \alpha_{\text{c}} S_{\text{clip}} + \alpha_{\text{t}} S_{\text{text}} + \alpha_{\text{r}} S_{\text{rel}}, \qquad \alpha_{\text{c}}, \alpha_{\text{t}}, \alpha_{\text{r}} \geq 0, \ \alpha_{\text{c}} + \alpha_{\text{t}} + \alpha_{\text{r}} = 1. \tag{36}$$

If relation cues are absent the weights are renormalized over the remaining terms.

**Relation features**  Let $B_g$ and $B_o$ denote the gripper and target boxes obtained using open vocabulary detection with a query set derived from $L$. Let $c(\cdot)$ return the box center and let $D$ be the image diagonal. Define normalized distance $d = \|c(B_g) - c(B_o)\|_2 / D$, intersection over union iou and a soft contact proxy $c^\star = \max\{\text{iou}, 1 - 3d\}$ clipped to $[0, 1]$. The vector $r$ is $[1 - d, \ \text{iou}, \ c^\star, \ \text{open}]$ with the last entry taken from robot telemetry if available.

D.4   Optional phase alignment from latent actions

When latent action tokens $A \in \mathbb{R}^{4 \times H}$ are recorded, a phase similarity can be computed by averaging across the four tokens and applying cosine

$$S_{\text{act}} = \tfrac{1}{2}\left(1 + \frac{\bar{a}_1^\top \bar{a}_2}{\|\bar{a}_1\|_2 \|\bar{a}_2\|_2}\right), \qquad \bar{a} = \tfrac{1}{4}\sum_{j=1}^{4} A_{j:}. \tag{37}$$

## D.5 Fusion strategies

The default additive fusion is

$$S = w_{\mathrm{e}}S_{\mathrm{embed}} + w_{\mathrm{i}}S_{\mathrm{image}} + w_{\mathrm{a}}S_{\mathrm{act}}, \qquad w_{\mathrm{e}}, w_{\mathrm{i}}, w_{\mathrm{a}} \geq 0, \ w_{\mathrm{e}} + w_{\mathrm{i}} + w_{\mathrm{a}} = 1. \tag{38}$$

In applications where the coarse descriptor must upper bound the final score one can prefer a gated composition

$$S = \big(\alpha + (1 - \alpha)S_{\mathrm{image}}\big)\, S_{\mathrm{embed}} + \beta\, S_{\mathrm{act}}, \tag{39}$$

with $\alpha \in [0,1]$ and a small $\beta$. This preserves identity on duplicate frames when $S_{\mathrm{embed}} \approx 1$ and allows the reranker to suppress false positives.

## D.6 Complexity and storage

With per step summary $(q, s)$ in dimension $d$, coarse retrieval requires a single dequantization and a cosine in $O(d)$. Large scale search uses an approximate nearest neighbor index on $\hat{u}$. The reranker is applied only to the top candidates and uses one forward pass of the vision language encoder and one run of open vocabulary detection when relation cues are enabled.

For a dataset with $N$ steps the storage for summaries is about $N \times (d + 4)$ bytes before container compression. With $N = 2 \cdot 10^5$ and $d = 256$ the raw size is around fifty two megabytes. This removes any need to retain the full token matrix.

## D.7 Why pooling across tokens and not across channels

Spatial tokens form a redundant set of local descriptors that must be aggregated into a single global representation. Channel directions span a learned semantic basis. Averaging across channels produces a length $T$ vector that measures token magnitudes while discarding the basis geometry. In contrast averaging across tokens preserves the channel geometry and leads to a descriptor that aligns with downstream cosine retrieval. Formally consider a linear probe $w \in \mathbb{R}^H$ that scores a semantic attribute. The pooled score from token pooling is $w^\top m$ which equals the mean of per token scores $w^\top \tilde{V}_{i:}$. Pooling across channels would instead collapse $w$ into a scalar and break linear separability.

## D.8 Pseudocode

---

**Algorithm 1** Step recording with compact descriptor

---

**Input** token features $V \in \mathbb{R}^{T \times H}$, image $I$, optional latent action $A$ and discrete action ids $g$; fixed projection $P \in \mathbb{R}^{H \times d}$.
**Output** image file and feature file containing $(q, s)$, optional $A, g$.

1: normalize tokens row wise to obtain $\tilde{V}$
2: compute $m$ and $x$ then $u_0$ as in the equations above
3: compute $u = \frac{u_0 P}{\|u_0 P\|_2 + \varepsilon}$
4: compute $s = \max_j |u_j|/127 + \delta$ and $q = \mathrm{round}(u/s)$
5: save $I$ and a compressed archive with $q$ as int8 and $s$ as float32 and optional $A, g$

---

**Algorithm 2** Language conditioned similarity

---

**Input** two observations $o_1, o_2$, instruction $L$
**Output** similarity $S \in [0, 1]$

1: dequantize stored summaries to obtain $\hat{u}_1, \hat{u}_2$ and compute $S_{\mathrm{embed}}$
2: encode images and text to obtain $z_1, z_2, t$ and compute $S_{\mathrm{clip}}, S_{\mathrm{text}}$
3: if relation cues are enabled then detect gripper and target from $L$, compute $r_1, r_2$ then $S_{\mathrm{rel}}$
4: combine into $S_{\mathrm{image}}$ using the weights $\alpha_{\mathrm{c}}, \alpha_{\mathrm{t}}, \alpha_{\mathrm{r}}$
5: if latent actions are present then compute $S_{\mathrm{act}}$
6: return fusion $S$ using either additive or gated composition

---

**Pairwise similarity**

## D.9 Recommended hyperparameters and practical notes

A projection dimension of two hundred fifty six offers a favorable accuracy storage tradeoff. The per vector quantizer with a single scale parameter is adequate for cosine based retrieval. The relation kernel uses standard deviations $\sigma$ on the order of one third of typical variation for each component. The instruction gate uses the minimum of image text cosines which is robust to asymmetric matches. The random projection is sampled once and stored, which ensures reproducibility across runs.

## D.10 Failure modes and mitigations

False positives can occur when appearance is similar but the instruction semantics differ. The instruction gate and the relation kernel address this issue. Failure due to detector misses can be mitigated by falling back to the combination of image cosine and text gate. Duplicate frames can be forced to score near one through a gated fusion that preserves the coarse score, and through identity short circuits when identical summaries are detected numerically.

## D.11 End to end retrieval

In large experience banks the proposed summary enables approximate nearest neighbor indexing. The query pipeline constructs the summary of the current frame, retrieves top candidates with $S_{\text{embed}}$, then applies the reranker on that shortlist and returns the final ordering. This design preserves instruction specificity while keeping the storage and time cost low.

# E Failure Case Analysis

We investigated reinforcement learning for EFN using the SAC algorithm, where the reward was defined by the semantic similarity between the predicted next observation and the reference observation sampled from the experience buffer. The Sinkhorn similarity metric was adopted to quantify semantic alignment in a high-dimensional embedding space. Despite its theoretical appeal, this design exhibited several flaws that ultimately caused training failure.

First, the reward distribution was too narrow. Denote the raw similarity at step $t$ as

$$R_t^{\text{raw}} = \text{SinkhornSim}\left(O_{t+1}, O_{t+1}^E\right), \tag{40}$$

where $O_{t+1}$ is the predicted next embedding and $O_{t+1}^E$ is the target embedding from the selected experience $E$. In practice, $R_t^{\text{raw}}$ remained tightly clustered in $[0.87, 0.93]$. As a result, the critic received only weak gradients because the variance

$$\text{Var}(R_t^{\text{raw}}) \approx 10^{-3} \tag{41}$$

was too small to differentiate between actions. To mitigate this, a normalization step was applied,

$$R_t^{\text{norm}} = \frac{R_t^{\text{raw}} - \mu_t}{\sigma_t}, \tag{42}$$

with running mean $\mu_t$ and variance $\sigma_t^2$. However, because $\mu_t$ was initialized near the empirical mean of the first batch, subsequent values of $R_t^{\text{raw}}$ were often less than $\mu_t$, producing predominantly negative $R_t^{\text{norm}}$. This inversion of sign transformed the optimization objective from maximizing positive returns to minimizing cumulative penalties, leading to unstable critic estimates and oscillatory Q-values.

Second, the target of imitation changed inconsistently across steps. At each time $t$, the experience trajectory $E$ was reselected to maximize semantic alignment with the current observation. This meant that the optimization target

$$\min_{\pi} \ \mathbb{E}\left[\ \|f(O_{t+1}) - f(O_{t+1}^E)\|^2\ \right] \tag{43}$$

was computed against a moving reference $O_{t+1}^E$ that varied with $t$. Frequent switching between different trajectories undermined the stationarity assumption of reinforcement learning and prevented

the critic from converging toward a consistent value function. The instability was exacerbated by the replay buffer, which stored transitions associated with outdated experience selections, further degrading learning.

Third, the scale of the embedding space posed intrinsic difficulties. Each observation embedding was represented as a tensor

$$O_t \in \mathbb{R}^{256 \times 4096}, \tag{44}$$

and the latent action correction operated in the space

$$\Delta A_t \in \mathbb{R}^{4 \times 4096}. \tag{45}$$

The dimensionality of these matrices is extremely high, with each step involving more than one million parameters in the observation representation alone. This created an environment where the actor–critic updates were easily overwhelmed by noise and variance, making exploration highly inefficient. SAC, which relies on accurate Q-function estimation, struggled to propagate meaningful gradients in such a vast search space.

In summary, the failure arose from a combination of (i) overly concentrated similarity-based rewards that became negative after normalization, (ii) non-stationary targets caused by continuous switching between different experience trajectories, and (iii) the excessive dimensionality of the embedding and action spaces that rendered stable credit assignment infeasible. This case highlights that reinforcement learning with high-dimensional semantic rewards requires careful reward shaping, fixed reference trajectories, and dimensionality reduction strategies in order to achieve stable convergence.

## F   Token-level Entropic Optimal Transport for SINKHORN_SIMILARITY

**Setting.**   Let $X \in \mathbb{R}^{T_x \times D}$ and $Y \in \mathbb{R}^{T_y \times D}$ denote token embeddings for two observations, with feature dimension $D$ and token counts $T_x$ and $T_y$. In our experiments $T_x = T_y = 256$ and $D = 4096$, while the derivation is general. The objective is to measure fine-grained agreement between $X$ and $Y$ without spatial pooling by aligning tokens through an entropic optimal transport plan.

**Cosine affinity.**   Each token is $\ell_2$–normalized along the feature dimension with a small numerical constant $\varepsilon_n > 0$:

$$\widehat{X}_{i\cdot} \;=\; \frac{X_{i\cdot}}{\|X_{i\cdot}\|_2 + \varepsilon_n}, \qquad \widehat{Y}_{j\cdot} \;=\; \frac{Y_{j\cdot}}{\|Y_{j\cdot}\|_2 + \varepsilon_n}. \tag{46}$$

The token-by-token cosine affinity is

$$S \;=\; \widehat{X}\widehat{Y}^{\top} \in [-1, 1]^{T_x \times T_y}, \qquad S_{ij} \;=\; \langle \widehat{X}_{i\cdot}, \widehat{Y}_{j\cdot} \rangle. \tag{47}$$

**Entropic OT objective.**   Set $C = -S$ so that larger similarity yields smaller cost. With uniform marginals

$$r \;=\; \tfrac{1}{T_x}\mathbf{1}_{T_x}, \qquad c \;=\; \tfrac{1}{T_y}\mathbf{1}_{T_y}, \tag{48}$$

the balanced entropic OT problem reads

$$\min_{P \in \mathbb{R}_{\geq 0}^{T_x \times T_y}} \; \langle P, C \rangle \;-\; \varepsilon\, H(P) \quad \text{subject to} \quad P\,\mathbf{1}_{T_y} = r, \;\; P^{\top}\mathbf{1}_{T_x} = c, \tag{49}$$

where $\varepsilon > 0$ controls the entropic regularization and

$$H(P) \;=\; -\sum_{i=1}^{T_x}\sum_{j=1}^{T_y} P_{ij}\big(\log P_{ij} - 1\big) \tag{50}$$

is the Shannon entropy. The entropy makes the problem strictly convex and yields a strictly positive solution.

**Gibbs kernel and Sinkhorn–Knopp scaling.** The optimal solution admits the factorization

$$P^\star = \text{diag}(u)\, K\, \text{diag}(v), \qquad K = \exp\left(\frac{S}{\varepsilon}\right), \tag{51}$$

for unique scaling vectors $u \in \mathbb{R}_{>0}^{T_x}$ and $v \in \mathbb{R}_{>0}^{T_y}$ that match the marginals. They are obtained by the fixed-point updates

$$\begin{aligned} u^{(t+1)} &= r \oslash \left(K\, v^{(t)} + \delta\right), \\ v^{(t+1)} &= c \oslash \left(K^\top u^{(t+1)} + \delta\right), \end{aligned} \tag{52}$$

where $\delta > 0$ stabilizes divisions and $\oslash$ denotes elementwise division. For strictly positive $K$ the iterations converge geometrically to the unique scaling pair.

**Similarity score and reward mapping.** The raw alignment value is the inner product between the transport plan and the affinity:

$$\text{score} = \langle P^\star, S \rangle = \sum_{i=1}^{T_x}\sum_{j=1}^{T_y} P_{ij}^\star S_{ij}. \tag{53}$$

Under uniform marginals one has

$$\sum_{i=1}^{T_x}\sum_{j=1}^{T_y} P_{ij}^\star = \mathbf{1}_{T_x}^\top P^\star \mathbf{1}_{T_y} = \mathbf{1}_{T_x}^\top r = 1, \tag{54}$$

hence score $\in [-1, 1]$ because $S_{ij} \in [-1, 1]$. For reinforcement learning the value is mapped to $[0, 1]$ as

$$\text{score}_{01} = \tfrac{1}{2}\left(\text{clip}(\text{score}, -1, 1) + 1\right). \tag{55}$$

---

**Algorithm 3** `sinkhorn_similarity` $(X, Y; \varepsilon, n_{\text{iters}})$

1: $\widehat{X} \leftarrow \text{row-normalize}(X)$, $\widehat{Y} \leftarrow \text{row-normalize}(Y)$
2: $S \leftarrow \widehat{X}\widehat{Y}^\top, \quad K \leftarrow \exp(S/\varepsilon)$
3: $r \leftarrow \frac{1}{T_x}\mathbf{1}_{T_x}, \quad c \leftarrow \frac{1}{T_y}\mathbf{1}_{T_y}$
4: $u \leftarrow \mathbf{1}_{T_x}, \quad v \leftarrow \mathbf{1}_{T_y}$
5: **for** $t = 1$ to $n_{\text{iters}}$ **do**
6: $\quad u \leftarrow r \oslash (Kv + \delta)$
7: $\quad v \leftarrow c \oslash (K^\top u + \delta)$
8: **end for**
9: $P \leftarrow \text{diag}(u)\, K\, \text{diag}(v)$
10: $\text{score} \leftarrow \langle P, S \rangle$
11: $\text{score}_{01} \leftarrow \frac{1}{2}\left(\text{clip}(\text{score}, -1, 1) + 1\right)$
12: **return** $\text{score}_{01}, P, S$

---

**Role of the temperature.** The parameter $\varepsilon$ governs the sharpness of the kernel. When $\varepsilon$ is small the kernel concentrates mass on pairs with large cosine similarity and the solution approaches a soft permutation that emphasizes near one-to-one matches. When $\varepsilon$ is large the plan spreads mass more diffusely and gradients become smoother. Empirically a range between $0.03$ and $0.1$ performs well for $T_x = T_y = 256$.

**Differentiability.** The map from $(X, Y)$ to $S$ is smooth after row normalization, the exponential kernel is smooth, and the Sinkhorn updates consist of matrix–vector products and stable elementwise operations. Gradients can be backpropagated through $u$ and $v$ into $K$ and $S$, and finally into $X$ and $Y$. If desired, the clipping in $\text{score}_{01}$ can be replaced by a smooth squashing such as a logistic function to avoid saturation.

**Complexity and memory.** For $T_x = T_y = T$, the kernel construction and each Sinkhorn iteration cost $O(T^2)$ time and $O(T^2)$ memory. With $T = 256$ the matrices contain 65,536 entries, which is tractable on modern accelerators even with multiple iterations.

**Numerical stabilization.**  Underflow and overflow can arise when $\varepsilon$ is very small.  A common remedy is recentering before exponentiation:

$$K \;=\; \exp\!\left(\frac{S - \max(S)}{\varepsilon}\right), \tag{56}$$

which leaves the optimal $P^\star$ unchanged up to rescaling of $u$ and $v$ because the Sinkhorn scaling absorbs global factors. The constant $\delta$ in the updates prevents division by zero in transiently sparse rows and columns.

**Relation to exact matching.**  Let $\Pi(r, c)$ denote the transport polytope with the given marginals. In the limit $\varepsilon \to 0$ one recovers the linear assignment problem

$$\arg\min_{P \in \Pi(r,c)} \langle P, -S \rangle, \tag{57}$$

whose solution becomes a permutation matrix when $T_x = T_y$ and the optimum is unique.  The entropic formulation therefore interpolates between hard assignment and a smooth strongly convex surrogate that is well suited to gradient-based learning.

**Reward semantics in EFN.**  The value $\text{score}_{01}$ quantifies token-accurate agreement between the predicted next observation and the target derived from the retrieved experience. Because mass can be distributed across multiple token correspondences according to their cosine affinity, the signal remains informative when only parts of the scene follow the intended dynamics, which reduces reward sparsity compared with global image pooling.

**Generalizations.**  Occlusions and appearance changes can be modeled by relaxing the marginal constraints with Kullback–Leibler penalties, leading to the unbalanced objective

$$\min_{P \geq 0} \; \langle P, C \rangle - \varepsilon H(P) + \tau_r \, \mathrm{KL}\!\left(P\mathbf{1}_{T_y} \,\|\, r\right) + \tau_c \, \mathrm{KL}\!\left(P^\top \mathbf{1}_{T_x} \,\|\, c\right), \tag{58}$$

which is compatible with generalized Sinkhorn updates. Nonuniform marginals can encode spatial priors by reweighting $r$ and $c$.

