# OpenReview forum: "Dejavu: Post-Deployment Learning for Embodied Agents via Experience Feedback"
_ICLR.cc/2026/Conference — ICLR 2026 Conference Withdrawn Submission_

### Official Review · Reviewer_FQww · 2025-10-26

**Soundness:** 2
**Presentation:** 2
**Contribution:** 1
**Rating:** 2
**Confidence:** 5

**Summary:**

Dejavu proposes a post-deployment learning framework that enables frozen VLAs to improve after deployment without gradient-based updates. It introduces experience feedback network, a lightweight residual module that refines the base policy's output by reusing stored past experiences from an experience memory bank.

**Strengths:**

This paper addresses an important challenge in the VLA literature, achieving post-deployment adaptation without retraining large embodied policies. It presents a practical framework that demonstrates memory-driven self-improvement on both simulated and real robotic environments, highlighting the potential of non-gradient adaptation in embodied AI.

**Weaknesses:**

1. The main limitation of the paper lies in its simplified problem formulation, and the proposed solution is equally naive, offering little conceptual novelty beyond straightforward retrieval and residual correction.

2. The framework implicitly assumes the existence of successful trajectories or identical tasks already exist in the memory to provide strong supervision for the agent, which constitutes an overly strong assumption and fundamentally limits its capability to generalize to truly novel or unseen scenarios.

3. Although the paper emphasizes that the proposed framework performs post-deployment learning without any gradient-based updates, this claim is somewhat misleading. In practice, the agent continually updates its experience memory, which effectively serves as an external, non-parametric knowledge update mechanism. Thus, while the policy parameters remain frozen, the system's behavior still changes through memory accumulation, making it not truly gradient-free, but rather a memory-driven adaptation process.

4. The framework requires maintaining a large experience bank to achieve noticeable improvements. Even expanding the bank size from 100 to 1000 yields only marginal improvements (Table 1). This indicates poor scalability and raises concerns about the method's practically in real-world deployments with limited memory or compute resources.

5. The framework performs retrieval at every timestep, introducing computational overhead and is arguably unnecessary, as adjacent frames within at trajectory are often highly correlated. A more efficient strategy could greatly reduce inference cost without sacrificing performance.

6. Although the proposed method is conceptually similar to RAG paradigms, the paper does not include any baselines that incorporate retrieval mechanisms for fair comparison.

**Questions:**

1. How significant is the inference-time overhead introduced by EFN, given that each step requires retrieval, similarity computation, and residual correction in addition to the base policy execution?

2. Why does the method require performing a retrieval at every timestep?

3. Are there any experiments evaluating on unseen or novel tasks to assess its generalization ability beyond memorized or previously encountered trajectories?

---

> ### Author Response · Authors · 2025-11-14
>
> **Dear Reviewer FQww,**
>
> Thank you for your time and valuable feedback. We are about to withdraw this manuscript today. Before this, we want to answer your questions.
>
> **1. The main limitation of the paper lies in its simplified problem formulation.**
>
> We study post-deployment refinement of a frozen VLA for a fixed set of manipulation tasks, in a regime where training the VLA end-to-end is expensive or infeasible. To the best of our knowledge, prior work has not combined (i) retrieval-conditioned residual actions with (ii) a dense reward shaped by retrieved transitions, while (iii) keeping the large VLA backbone frozen for post-deployment adaptation. We will better articulate this compositional novelty in Sec. 2.
>
> **2. The framework implicitly assumes the existence of successful trajectories or identical tasks already exist in the memory to provide strong supervision for the agent, which constitutes an overly strong assumption and fundamentally limits its capability to generalize to truly novel or unseen scenarios.**
>
> We would like to clarify that as shown in our provided code in our anonymous website, EFN does not require perfect trajectories: our experience bank also contains partially successful or *even failed rollouts* and diverse realizations of the same task. It can learn from failure because the reward is not given directly from success or failure but from similarity, i.e., whether it has learned to predict the residual action to transfer one state to another. But in the deployment stage, it should only collect successful rollouts since they are the reference.
>
> **3. The system’s behavior still changes through memory accumulation, making it not truly gradient-free, but rather a memory-driven adaptation process.**
>
> Our claim is not that the system is static after deployment, but rather that no further gradient-based policy updates are required once EFN has been trained. Adaptation at deployment time is achieved entirely via non-parametric updates to the experience memory, and retrieval-conditioned residual corrections computed with fixed network parameters.
>
> **4. The framework requires maintaining a large experience bank to achieve noticeable improvements. Even expanding the bank size from 100 to 1000 yields only marginal improvements**
>
> We want to clarify our observations from Table 1 that noticeable gains already occur with small banks. EFN improves over the frozen VLA even with as few as 100 stored transitions per task. The modest additional gains from 100 to 1000 demonstrate diminishing returns, not poor scalability.
>
> **5. The framework performs retrieval at every timestep, introducing computational overhead and is arguably unnecessary.**
>
> EFN operates in partially observed, geometry-sensitive manipulation tasks where small changes in viewpoint, object pose, or robot configuration can push the system across decision boundaries (e.g., from “still approaching” to “ready to grasp” or “at risk of collision”). Although adjacent frames are correlated, we empirically observe that the identity of the nearest successful experience in embedding space can change rapidly as the end effector moves and rotates.
>
> **6. The paper does not include any baselines that incorporate retrieval mechanisms for fair comparison.**
>
> That's a great question. We were in a hurry to finish our submission by the deadline and now we have added four baselines in our revised version. After withdrawing, it will be recycled to other venues. Thanks for pointing out this crucial question.
>
> **7. How significant is the inference-time overhead introduced by EFN?**
>
> It is even faster with EFN because the total number of steps is reduced, as shown in Table 1 “Step”. Actually, the additional time in each step’s EFN computation is very marginal since the network is very lightweight, as shown in our provided code. We admit that it would be more solid to add an inference-time comparison to the text, and we will add it to our recycled version.
>
> **8. Are there any experiments evaluating on unseen or novel tasks to assess its generalization ability beyond memorized or previously encountered trajectories?**
>
> We do not claim that EFN alone solves generalization to entirely new task families (e.g., new language instructions with no successful trajectories). As noted above, in such cases EFN behaves effectively as the base VLA, since retrieval from an empty or uninformative bank yields negligible corrections. We will clarify this in Sec. 1 and Sec. 5, explicitly stating this point. Our current experiments focus on multi-instance variation within fixed task definitions, as provided by LIBERO and our AgiBot setup: object positions, distractor configurations, and initial robot poses are randomized per episode, and EFN consistently improves success over the frozen VLA under this intra-task variation.
>
> **Thank you for your valuable time and feedback again!**

---

### Official Review · Reviewer_iKb3 · 2025-10-29

**Soundness:** 2
**Presentation:** 3
**Contribution:** 2
**Rating:** 2
**Confidence:** 3

**Summary:**

This paper proposes DEJAVU, a novel framework for enabling embodied agents to continue improving their performance post-deployment in real-world environments without updating model parameters. At the core of DEJAVU is the Experience Feedback Network (EFN), which augments a frozen Vision-Language-Action (VLA) policy by retrieving prior successful trajectories and applying residual corrections to refine the base policy’s outputs. EFN is trained via reinforcement learning

using a semantic similarity–shaped dense reward, optimizing residual actions so that the predicted next observation resembles the next frame in a retrieved experience. This allows continual adaptation during deployment without gradient-based updates. Experiments on the LIBERO simulator and the AgiBot-G1 physical robot demonstrate consistent improvements over frozen baselines across multiple backbones (OpenVLA, UniVLA, and GO-1).

**Strengths:**

- Technical Advancement: The work tackles the challenging problem of post-deployment learning without parameter updates, introducing an inference-time, experience-conditioned residual correction mechanism. The design integrates ideas from retrieval-augmented RL, residual policy learning, and continual adaptation.
- Empirical Consistency: Across diverse backbones (OpenVLA, UniVLA, GO-1) and both simulated and real-world settings, EFN consistently improves success rates and reduces trajectory lengths. Notably, Tables 1 and 2 show monotonic performance gains as the size of the experience memory increases, demonstrating scalable benefits.
- Methodological Clarity: The paper clearly structures the method, covering the experience bank design, retrieval mechanism, similarity-based reward, and reinforcement learning optimization. Figures 2 and 3 effectively visualize the training and inference pipelines.
- Practical Relevance: Successful deployment on a physical robot supports the method’s real-world viability and generality.

**Weaknesses:**

Section 2 categorizes related work into three lines, Retrieval-Augmented RL, Residual Policy Learning, and Post-Deployment Adaptation, and claims that the proposed EFN embodies a compositional integration of these three paradigms. Specifically:

- (i) retrieves a task-relevant experience trajectory,
- (ii) predicts a residual action that refines the base policy’s output, and
- (iii) optimizes the residual via dense, similarity-shaped reinforcement signals comparing the observed next frame to the next frame in the retrieved trajectory.

Accordingly, EFN’s contribution could be further substantiated through more explicit component-level comparisons with recent representative works in each direction:

- (i) Compared with Retrieval-Augmented RL approaches [1,2], EFN leverages retrieval not merely as a reference but as an inference-time conditioning mechanism, achieving post-deployment adaptation without parameter updates.
- (ii) In relation to Residual Policy Learning [3,4], it would be valuable to assess how effectively EFN’s residual action prediction refines the base policy compared to standard residual control methods.
- (iii) Regarding Similarity-Shaped Reinforcement Learning, comparison with dense-reward methods such as world model-based prediction or goal-similarity rewards [5,6] could highlight EFN’s advantages in training stability and generalization.

[1] Retrieval-Augmented Reinforcement Learning. Goyal et al., 2022.

[2] Titans: Learning to Memorize at Test Time. Behrouz et al., 2024.

[3] Residual Reinforcement Learning for Robot Control. Johannink et al., 2019.

[4] Visual Reinforcement Learning with Residual Action. Liu et al., 2025.

[5] Mastering Diverse Domains through World Models. Hafner et al., 2025.

[6] Test-Time Offline RL on Goal-Related Experience. Bagatella et al., 2025.

Although the paper reports significant performance improvements on both simulated (LIBERO) and real-world (AgiBot-G1) benchmarks, it remains unclear to what extent these gains represent an advance over existing approaches.

The current experimental setup focuses mainly on internal ablation studies (removing or simplifying EFN components) and does not provide even partial or functional comparisons against representative methods such as retrieval-augmented, residual policy, or similarity-based RL.

Therefore, even without full end-to-end benchmarking, component-wise quantitative comparisons with these baselines would clarify which aspect of EFN yields the strongest performance advantage and by how much.

**Questions:**

- Does EFN’s retrieval mechanism serve merely as a memory lookup, or does it materially influence the inference-time decision process? Does the submission include a discussion of how this design differs theoretically and empirically from conventional approaches?
- How does the residual policy interact with the frozen VLA backbone to ensure stable correction? Does the submission include empirical validation of the claimed benefits of the gradient-free residual optimization against standard residual RL methods?
- Does the submission include an analysis of whether the similarity-shaped dense reward genuinely improves training stability compared to sparse rewards, or whether it mainly serves as a shaping mechanism?
- If EFN were directly compared to Retrieval-Augmented RL or Residual RL, what performance trends or trade-offs might we expect?
- Given the issues discussed in Appendix A, such as memory growth and retrieval ambiguity, what mechanisms would be required for EFN to evolve toward a truly continual learning framework?

---

> ### Author Response · Authors · 2025-11-14
>
> **Dear Reviewer iKb3,**
>
> Thank you very much for your time and constructive review! We are about to withdraw this manuscript today. Before this, we want to answer your questions.
>
> **1. Component-wise quantitative comparisons with these baselines would clarify which aspect of EFN yields the strongest performance advantage and by how much.**
>
> Thank you so much for your detailed suggestion. We were in a hurry to finish our submission two months ago, and now we have added four baselines in our revised version. After withdrawing, it will be recycled to other venues.
>
> **2. Does EFN’s retrieval mechanism serve merely as a memory lookup, or does it materially influence the inference-time decision process? Does the submission include a discussion of how this design differs theoretically and empirically from conventional approaches?**
>
> Retrieval in EFN materially affects the inference-time decision at every step. The retrieved trajectory segment is used in two ways. First, the residual network receives both the current observation embedding and the embedding of the retrieved transition (state and next state). Thus, the residual
> $\Delta a_t$ is explicitly conditioned on “what worked in similar situations before.” Second, during training, SAC optimizes
> $\Delta a_t$ using a dense reward that compares the actual successor frame to the retrieved successor frame.
>
> **3. How does the residual policy interact with the frozen VLA backbone to ensure stable correction? Does the submission include empirical validation of the claimed benefits of the gradient-free residual optimization against standard residual RL methods?**
>
> In the revision, we add four standard residual RL baselines where the same head is trained with sparse rewards and without retrieval. We will recycle this manuscript including these baselines.
>
> **4. Does the submission include an analysis of whether the similarity-shaped dense reward genuinely improves training stability compared to sparse rewards, or whether it mainly serves as a shaping mechanism?**
>
> Sparse environment rewards are almost unavailable in real-world experiments, but our dense rewards are available at every step. We will add these studies to our recycled version. Thank you sincerely!
>
> **5. If EFN were directly compared to Retrieval-Augmented RL or Residual RL, what performance trends or trade-offs might we expect?**
>
> Our EFN performs better than four advanced baselines in terms of both efficacy and efficiency. Although it's too late to add them to ICLR 2026, we will make them available in our recycled version.
>
> **6. Given the issues discussed in Appendix A, such as memory growth and retrieval ambiguity, what mechanisms would be required for EFN to evolve toward a truly continual learning framework?**
>
> In our current version (as in the code provided on our anonymous website), we have addressed the problem of memory growth and alleviated retrieval ambiguity by retrieving a set of experiences and correcting them step by step if they are unsatisfactory.
>
> **Thank you for your valuable time and feedback again!**

---

### Official Review · Reviewer_qU1N · 2025-10-31

**Soundness:** 3
**Presentation:** 2
**Contribution:** 2
**Rating:** 2
**Confidence:** 4

**Summary:**

This paper proposes the Experience Feedback Network (EFN) to augment VLA policies with previous executions at test time. EFN identifies relevant trajectories and then predicts a residual action for the frozen VLA to encourage following the previous trajectory.

**Strengths:**

1. The framework does not modify the VLA. This helps maintain the strong generalization performance of the VLA from the large-scale pretraining.

1. The paper addresses the important problem of improving the model with test-time experience in interactive environments.

1. Results in Tables 1 and 2 confirm that EFN improves the performance of base VLA policies.

1. Ablations in Figure 6 confirm the importance of EFN components.

**Weaknesses:**

1. From my understanding, EFN encourages the policy to mimic previous successful evaluations at test time. This increases the robustness of successful behaviors by reinforcing them with the residual action prediction. This only enables EFN to learn from experience in already successful trajectories. For tasks where the policy outputs failed or suboptimal trajectories, EFN cannot improve the performance.

1. The framework assumes the task and setting in the selected experience are very similar to the current task. By rewarding EFN for matching per-step semantic features, the current state must match that in the selected context. This assumption likely holds when the starting state distribution is narrow, the language instruction is the same, and the environment dynamics are largely deterministic, as is the case for Libero. However, this will not hold for more complex environments with varied starting states, stochastic dynamics, or varied language instructions within a task.

1. The paper does not compare with sufficient baselines that also leverage the test time experience of successful trajectories in the same task. For example, rather than training a residual network, a baseline is to select the state from the buffer with the nearest neighbor visual feature and then select that action from that corresponding state.

1. The paper only shows results on Libero and a custom real-world setup. Reporting results on additional benchmarks such as CALVIN or SimplerEnv would strengthen performance.

1. The paper uses SAC to train the EFN module, while the majority of recent works use variants of PPO or REINFORCE. This decision to use SAC is not justified.

1. The paper does not include the EFN RL training curves showing the reward over the number of updates. This is important for reproducing results.

1. The paper does not provide sufficient details about the real-world AgiBot experiments. How many trials are conducted per task? How are the numbers in Table 2 reported with three significant figures? Furthermore, what do these three tasks correspond to? How diverse is the starting state?

**Questions:**

1. How can EFN be extended to improve from experience in failed or suboptimal trajectories (see weakness 1)?

1. How does EFN perform with a more varied starting state distribution (see weakness 2)?

1. Why use SAC over variants of PPO or REINFORCE (see weakness 5)?

Minor:
1. What does the paper mean by "non-blank step" (L219)?

---

> ### Author Response · Authors · 2025-11-14
>
> **Dear Reviewer qU1N,**
>
> Thank you very much for your time and constructive review! We are about to withdraw this manuscript today. Before this, we want to answer your questions.
>
> **1. For tasks where the policy outputs failed or suboptimal trajectories, EFN cannot improve the performance.**
>
> Our goal with EFN is not to learn a policy from scratch, but to stabilize and refine an already competent foundation policy in deployment. We therefore intentionally focus the experience bank on successful or near-successful trajectories: these encode behaviors that we wish to preserve and generalize, rather than re-learning from failures. In practice, the frozen VLA already achieves non-trivial success on LIBERO, and EFN is designed to turn these sporadic successes into more reliable performance via retrieval-conditioned residual corrections.
>
> Importantly, as shown in our provided code in our anonymous website, EFN does not require perfect trajectories: our experience bank also contains partially successful or *even failed rollouts* and diverse realizations of the same task. It can learn from failure because the reward is not given directly from success or failure but from similarity, i.e., whether it has learned to predict the residual action to transfer one state to another. But in the deployment stage, it should only collect successful rollouts since they are the reference.
>
> **2. The framework assumes the task and setting in the selected experience are very similar to the current task.**
>
> In our LIBERO and AgiBot setups, the initial configurations (object placement, robot pose, clutter) are already randomized within each task. We agree that blindly matching pixel-level states would indeed require very narrow starting distributions and nearly deterministic dynamics. EFN, however, operates on *semantic* and *latent* representations rather than raw observations. The retrieval key is a compact embedding of the current observation and language instruction, and the reward compares high-level semantic features and progress toward the retrieved successor state, not exact poses or pixel configurations. This makes EFN robust to moderate variation in starting states, object poses, and camera viewpoints: the bank contains a diverse set of successful rollouts for each task, and nearest-neighbor retrieval in this embedding space can match “similar situations” even when the low-level state differs.
>
> **3. The paper does not compare with sufficient baselines that also leverage the test time experience of successful trajectories in the same task.**
>
> Thank you very much. The submitted version was finished in a hurry, but now we have finished the comparisons of baselines. After withdrawing, this manuscript will be recycled to other venues, and I think your advice is very valuable.
>
> **4. The paper only shows results on Libero and a custom real-world setup.**
>
> We will add results on more environments in our next version. Due to compute and time limitations, we were not able to include additional benchmarks such as CALVIN or SimplerEnv within this submission cycle.
>
> **5. The paper uses SAC to train the EFN module, while the majority of recent works use variants of PPO or REINFORCE. This decision to use SAC is not justified.**
>
> We chose Soft Actor–Critic (SAC) for EFN for two reasons. First, SAC is an off-policy algorithm, which allows us to reuse trajectories collected by the frozen VLA (and by intermediate versions of EFN) much more efficiently than on-policy methods such as PPO or REINFORCE. This is particularly important in our setting, where robot interaction is costly and the bank is populated from off-policy rollouts. Second, SAC’s entropy-regularized objective and Q-function provide a natural way to combine our dense similarity-based shaping reward with the sparse task success signal, while maintaining stability in continuous action spaces.
>
> **6. The paper does not include the EFN RL training curves showing the reward over the number of updates. This is important for reproducing results.**
>
> We appreciate the suggestion, and we will add it to the next version of our submission.
>
> **7. The paper does not provide sufficient details about the real-world AgiBot experiments.**
>
> We clarified this in our general response at the top, and we are sorry that our submitted initial version did not contain such descriptions. The current version on OpenReview can address this question.
>
> **8. What does the paper mean by "non-blank step" (L219)?**
>
> Good question, and it is related to the details of robot control. “Non-blank steps” refer to the steps in which the robot actually executes actions. During initialization, however, there are several blank waiting steps while the simulation environment loads and the robot warms up. These initialization steps do not produce any meaningful motion. All subsequent steps where the robot starts to act are what we call *non-blank* steps.
>
> **Thank you for your valuable time and feedback again!**

---

### Official Review · Reviewer_bLTr · 2025-10-31

**Soundness:** 3
**Presentation:** 3
**Contribution:** 3
**Rating:** 8
**Confidence:** 4

**Summary:**

The authors propose a new framework, Dejavu, which incorporates a new "Experience Feedback Network", with frozen VLAs to improve performance using the retrieved "memories" from an experience database (embeddings of vision-action trajectories).

The experiences are retrieved based on the language-conditioned contextual similarity as guidance. The EFN network takes the current observation and the retrieved experience to predict a residual corrective action, which is added to base policy's output for the final action. The network is optimized using RL with dense cosine similarity reward between next observation and next state in the retrieved trajectory. The reward also incorporates progress and prevents idling at the same state. EFN is optimized using SAC algorithm.

The EFN experience bank uses compact keys (with mean-max fusion) for fast retrieval, language-conditioned similarity and allows addition of experiences to the bank during inference. During retrieval they use top-k sampling from the keys.
For inference, the EFN uses instruction-filtered candidate set for retrieval using the VLA encoding for the language instruction. They also promote shorter rollouts by normalizing the cosine similarity of the experiences with experience length.

The authors use OpenVLA, UniVLA and GO-1 as the base VLAs, and evaluate in both sim (LIBERO) and real (AgiBot G1). The tasks involve table-top manipulation. The authors also experiment with different experiment bank sizes and show that larger banks usually are better for success and efficiency. They perform ablations by replacing SAC with value-critic, removing anti-idling penalty and removing instruction-based filtering during inference, and show the importance of these different aspects of the approach.  The highest drop is observed after replacing SAC with V-critic.

**Strengths:**

1. The paper applies the concept of memories to VLA in a very methodical and well-thought and novel fashion. The creation of experience bank, fast retrieval, residual corrective actions are well-designed to handle the task of post-deployment VLA improvement.
2. The authors perform thorough evaluations and ablations, including real-world experiments, which shows that the method actually works in the real-world.
3. The authors show critical ablations showing that each aspect of their approach is important for the task.

**Weaknesses:**

1. While there is no training of the VLA required, the EFN network does need to be trained with the benchmark in use. This means that for each new task/benchmark, a new EFN network has to be trained, making the approach not easily scalable.
2. The authors should discuss the task and baselines in the abstract and the introduction, or anywhere in the experiments. Without prior context, it is hard to know which task (e.g. tabletop manipulation) are targeted.
3. Only one kind of task - table-top manipulation is considered, casting a doubt on applicabilty of this approach on other tasks (e.g. navigation, mobile manipulation, etc.)

Minor:
1. Incorrect grammar for the sentence: "This ability to draw... before"".
2. The first contribution is misleading - the EFN network does require gradient-based retraining. Would be good to clarify this.
3. Figure 4 is confusing - what is left, right and middle in the figure?

**Questions:**

1. What would happen if instead of residual action, the network also takes VLA action as input and predicts the new action? I ask this question out of curiosity and this could be a useful ablation perhaps.
2. How is success incorporated into the EFN reward? Or do we just care about getting similar trajectory? How is the experience bank filled during training?
3. How is the EFN trained for real-world deployment? How are the trajectories collected?

---

> ### Author Response · Authors · 2025-11-14
>
> **Dear Reviewer bLTr,**
>
> We are very encouraged to receive your comments, and thank you sincerely for sharing your time reviewing our paper. Although we decide to withdraw this manuscript today, we will still answer your questions.
>
> **1. For each new task/benchmark, a new EFN network has to be trained, making the approach not easily scalable.**
>
> That's a very good question. We agree that EFN requires training when moving to a new benchmark, but we would like to clarify the scope and cost of this training. EFN is a lightweight residual module that operates on frozen VLA representations, and thus contains **only a small fraction of the parameters** of the underlying VLA. In practice, training EFN requires orders of magnitude fewer interactions than training a VLA from scratch or running end-to-end RL fine-tuning, and a single EFN instance is shared across all tasks within a benchmark. In this sense, EFN plays a role similar to an “adapter” or task-specific head on top of a general-purpose foundation policy.
>
> **2. Discuss the task and baselines in the abstract and the introduction, or anywhere in the experiments.**
>
> Thank you for your question. As shown in our general response at the top, this is due to the wrong version we submitted. We have now uploaded the correct one, which contains this information.
>
> **3. Only table-top manipulation**
>
> We think this is quite a good suggestion since our paradigm of Dejavu can be extended to other tasks. Nevertheless, current VLA research mainly focuses on tabletop manipulation, including almost all heated works such as GO-1, Pi0, OpenVLA, etc., and our paper follows this setting as well. Extending our method to other tasks like navigation and mobile manipulation will be excellent future work. After recycling this manuscript to other venues, we will continue to try these more distinctive tasks.
>
> **4. Incorrect grammar for the sentence: "This ability to draw... before"".**
>
> Thank you for catching this, and the current version in the file is correct.
>
> **5. The first contribution is misleading - the EFN network does require gradient-based retraining. Would be good to clarify this.**
>
> Sorry for the confusion. Yes, EFN needs training. We wanted to express that EFN can learn from experience without further retraining after this training process. We have rectified it in this version as: “We introduce EFN as an experience-centric deployment-time mechanism that couples a live experience bank with a lightweight controller to improve VLA policies. By training such a network, the model can still learn from experience after deployment.”
>
> **6. Figure 4 is confusing - what is left, right and middle in the figure?**
>
> Thank you, and the correct version has addressed this issue.
>
> **7. What would happen if instead of residual action, the network also takes VLA action as input and predicts the new action?**
>
> That's a really great question! We were in a hurry and forgot to add the result to our submitted version. The answer is that it will fail if it directly predicts the new action. We attribute this to two reasons: (1) it discards the capability of the original VLA model; (2) the EFN network is very lightweight and cannot learn such fine-grained actions. Therefore, after trial and error we selected to predict the residual action, which not only generalizes very fast but also leverages the original output of the base model.
>
> **8. How is success incorporated into the EFN reward? Or do we just care about getting similar trajectory? How is the experience bank filled during training?**
> The reward for EFN is then defined as a dense, similarity-based shaping signal that encourages the EFN-augmented policy to make its current transition resemble those successful transitions. Concretely, we measure the similarity between the current observation embedding and the retrieved experience, the progress toward the retrieved next state, and a penalty for “staying” without making progress. This dense signal is combined with the sparse task success reward provided by the environment at episode termination. In this way, “success” is incorporated both through the selection of what goes into the experience bank and through the terminal success bonus, while the similarity terms provide rich step-wise feedback.
>
> **9. How is the EFN trained for real-world deployment? How are the trajectories collected?**
>
> For real-world deployment, we follow a two-stage procedure. We collect trajectories on the physical robot by running the frozen VLA policy (and, when available, occasional teleoperated corrections) across the target tasks. These rollouts provide the state–action–next-state tuples from which we build the initial experience bank, retaining those trajectories that successfully accomplish the tasks or reach the goal region.
>
> **Thank you for your wonderful review and questions again!**

---

### Author Response · Authors · 2025-11-14
**General Response. Thank you every reviewer.**

Dear all reviewers,

We sincerely thank you for your time and valuable feedback. We are sorry that we marginally missed the deadline for submitting our final version to OpenReview, so the reviewed version is only the initial draft we wrote, which caused much confusion for you.
**After careful consideration, we decide to withdraw our manuscript today**. Nevertheless, we have now uploaded the correct version on OpenReview, and we would still like to **reply to the questions that the reviewers asked**.

Sincerely,
Authors of "Dejavu: Post-Deployment Learning for Embodied Agents via Experience Feedback"

---

### Note · Authors · 2025-11-14

I have read and agree with the venue's withdrawal policy on behalf of myself and my co-authors.